# LeFusion: Controllable Pathology Synthesis via Lesion-Focused Diffusion Models

**Hantao Zhang**[1,2]*, **Yuhe Liu**[3], **Jiancheng Yang**[1]†, **Shouhong Wan**[2],
**Xinyuan Wang**[3], **Wei Peng**[4], **Pascal Fua**[1]
[1]Swiss Federal Institute of Technology Lausanne (EPFL), Switzerland
[2]University of Science and Technology of China (USTC), China
[3]Beihang University, China
[4]Stanford University, USA

## Abstract

Patient data from real-world clinical practice often suffers from data scarcity and long-tail imbalances, leading to biased outcomes or algorithmic unfairness. This study addresses these challenges by generating lesion-containing image-segmentation pairs from lesion-free images. Previous efforts in medical imaging synthesis have struggled with separating lesion information from background, resulting in low-quality backgrounds and limited control over the synthetic output. Inspired by diffusion-based image inpainting, we propose LeFusion, a lesion-focused diffusion model. By redesigning the diffusion learning objectives to focus on lesion areas, we simplify the learning process and improve control over the output while preserving high-fidelity backgrounds by integrating forward-diffused background contexts into the reverse diffusion process. Additionally, we tackle two major challenges in lesion texture synthesis: 1) multi-peak and 2) multi-class lesions. We introduce two effective strategies: histogram-based texture control and multi-channel decomposition, enabling the controlled generation of high-quality lesions in difficult scenarios. Furthermore, we incorporate lesion mask diffusion, allowing control over lesion size, location, and boundary, thus increasing lesion diversity. Validated on 3D cardiac lesion MRI and lung nodule CT datasets, LeFusion-generated data significantly improves the performance of state-of-the-art segmentation models, including nnUNet and SwinUNETR. Code and model are available at `https://github.com/M3DV/LeFusion`.

## 1 Introduction

The development of AI for healthcare often suffers from data scarcity (Ibrahim et al., 2021; Schäfer et al., 2024). In most biomedical scenarios, the number of pathological subjects is significantly lower than that of normal ones. This discrepancy primarily arises from the naturally occurring distribution of patient data, which frequently exhibits long-tail characteristics (Yang et al., 2022; Zhang et al., 2023). Additionally, potential biases in data collection can introduce issues related to algorithmic fairness (Xu et al., 2022; Chen et al., 2023; Yang et al., 2024), as well as concerns about security and privacy (Price & Cohen, 2019; Qayyum et al., 2020). As a result, it has been argued that "synthetic data can be better than real data" (Savage, 2023).

Generative lesion synthesis is a promising approach to generating diverse medical data, benefiting many medical applications (Khader et al., 2023). By learning from lesion-containing data, generative models can synthesize various types of lesions, which in turn benefit downstream applications (Han et al., 2019; Jin et al., 2021; Yang et al., 2019; Lyu et al., 2022a; Shin et al., 2018; Pishva et al., 2023; Du et al., 2023; Lyu et al., 2022b). While a range of generative methods have been explored, they often struggle to preserve high-quality backgrounds outside of the lesion areas. This is because generating anatomically correct backgrounds in the human body is far more challenging than synthesizing isolated lesions. Moreover, these methods often lack control over key

---

*This work was conducted during the first author's research internship at EPFL.
†Corresponding author: Jiancheng Yang (`jiancheng.yang@epfl.ch`), who led the project.

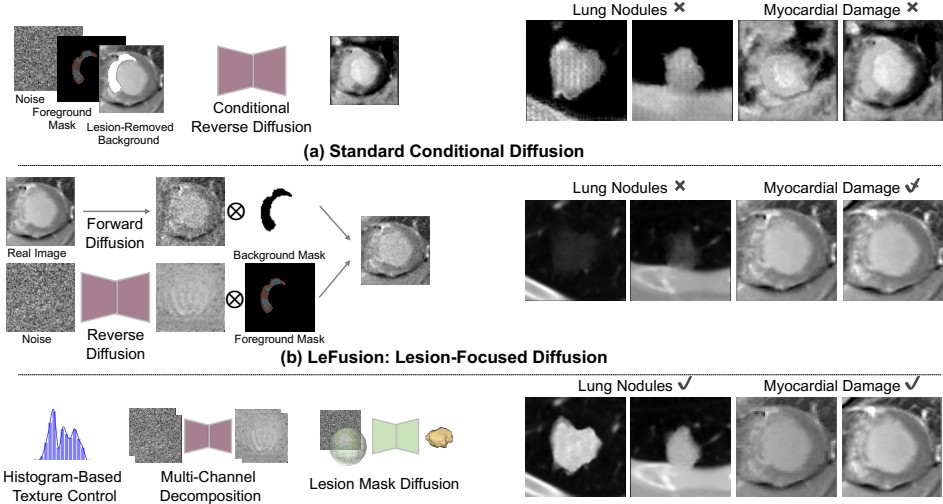

Figure 1: **Standard Conditional Diffusion vs. Lesion-Focused Diffusion (LeFusion).** (a) Standard Conditional Diffusion concatenates background, lesion mask and noise as inputs and outputs both lesion and background, risking background integrity and wasting capacity on difficult but unnecessary background generation, especially in data-limited settings. (b) LeFusion uses forward-diffused backgrounds and reverse-diffused foregrounds as input. The model generates only the lesion, ensuring realistic background preservation and simplifying the task. (c) LeFusion with Fine Control of Lesion Textures and Masks introduces histogram-based texture control for multi-peak lesions (lung nodules), multi-channel decomposition for multi-class lesions (myocardial damage), and lesion mask diffusion for control over size, location and boundary, enhancing quality and diversity.

aspects of lesion generation, including texture type, size, location, and mask alignment. These issues can severely degrade the performance of downstream applications, such as segmentation algorithms trained using such synthetic data. Fig. 1 (a) illustrates standard diffusion models as an example.

One approach to avoiding the complexity of background generation is to start from readily available normal scans and synthesize lesions into them. This involves generating lesion masks and filling them with appropriate textures, ensuring perfect background preservation and mask alignment, while also allowing precise control over lesion size and location. In this paper, we refer to this method as background-preserving lesion synthesis. This approach has led to a resurgence of hand-crafted methods, which have been used to model COVID-19 lesions (Yao et al., 2021) and liver tumors (Hu et al., 2023). However, these methods rely on heuristics that do not generalize well.

Inspired by diffusion-based image inpainting schemes (Lugmayr et al., 2022; Avrahami et al., 2022), it has been shown that explicitly integrating real background context during the diffusion process ensures realistic background preservation outside lesion masks. Rather than using the background as conditional inputs (Rombach et al., 2022), these methods directly incorporate forward-diffused background contexts into the reverse diffusion process. However, these approaches are training-free and do not focus on specific inpainted content. In contrast, our study emphasizes lesion generation. In data-limited scenarios, it is more efficient for the model to focus solely on lesion synthesis.

To this end, we propose *LeFusion*, a lesion-focused diffusion model (Fig. 1b and Fig. 1c). We redesign the diffusion learning objectives to focus solely on lesion data. Similar to diffusion-based inpainting, the input combines forward-diffused backgrounds with reverse-diffused foregrounds, while the model reconstructs only the lesion, avoiding the need to allocate capacity to learning complex backgrounds. Furthermore, we address two major unresolved challenges in lesion synthesis: 1) multi-peak lesions, where lesions have distinct types, and 2) multi-class lesions, where multiple classes of lesions need to be generated simultaneously. For the multi-peak challenge, we introduce histogram-based texture control, integrating lesion texture histograms during training as a condition, which allows control over lesion types during inference. We find that explicitly controlling the histogram is crucial when generating lesions on normal scans; otherwise, the model tends to produce lesions biased toward healthy appearances. Notably, this histogram-based control method is generic and does not require any additional information beyond image-mask pairs. To handle the

multi-class challenge, we propose a strategy for joint modeling of multi-class lesions through multi-channel decomposition, where the diffusion model generates different lesion classes via separate channels and then combines them into a single image. Finally, we introduce lesion mask diffusion, enabling control over size, location and boundary, thereby increasing the diversity of lesion masks.

We validate LeFusion on 3D lung nodule CT datasets (Armato III et al., 2011) and cardiac lesion MRI (Lalande et al., 2022), demonstrating its effectiveness in addressing both the multi-peak and multi-class challenges while generating high-quality synthetic lesions. In downstream segmentation tasks, we show that LeFusion-generated data significantly enhances the performance of state-of-the-art models such as nnUNet (Isensee et al., 2021) and SwinUNETR (Hatamizadeh et al., 2021).

## 2 RELATED WORK

### 2.1 GENERATIVE MODELS FOR LESION SYNTHESIS

Lesion synthesis using generative models has garnered significant attention for its potential to create diverse and realistic medical datasets, particularly in addressing the scarcity and imbalance of pathological data in biomedical applications. Early approaches have employed variational autoencoders (VAEs)(Kingma & Welling, 2013), generative adversarial networks (GANs)(Goodfellow et al., 2020), and more recently, diffusion models (Ho et al., 2020) to generate synthetic lesions across various medical imaging modalities and applications, including lung nodules (Han et al., 2019; Jin et al., 2021; Yang et al., 2019) and COVID-19 lesions (Lyu et al., 2022a) in CT scans, colon polyps in colonoscopy (Shin et al., 2018; Pishva et al., 2023; Du et al., 2023), tumor cells in microscopy (Horvath et al., 2022), brain and brain lesions in MRI (Billot et al., 2023), diabetic lesions in retinal images (Wang et al., 2022), and synthetic liver tumors (Lyu et al., 2022b).

However, a persistent challenge for these methods is preserving anatomically accurate backgrounds alongside the lesions. In medical imaging, the background must respect the anatomical structure of the human body, which makes generating realistic backgrounds significantly more difficult than synthesizing isolated lesions, particularly in large-scale 3D images. While recent studies (Hamamci et al., 2024; Peng et al., 2024) have begun to address large-scale 3D medical image generation, these methods often require significant computational resources and extensive data.

Another limitation of current methods is the lack of explicit control when generating image-mask pairs. Typically, both the lesion and its corresponding mask are generated simultaneously, without explicit constraints linking the two. This results in limited control over key lesion properties, such as texture, size, location, and alignment between the image and mask. The absence of high-quality background preservation and fine control over these properties hinders the scalability and effectiveness, negatively impacting the performance of downstream tasks such as segmentation.

### 2.2 BACKGROUND-PRESERVING LESION SYNTHESIS

In clinical practice, normal scans (either whole or partial) are far more abundant than pathological ones. For instance, in lung nodule cases, most pathological scans contain only a single lesion, yet traditional lesion synthesis methods often focus on small crops around the lesion, utilizing as little as $< 1\%$[1] of the original data and leaving large portions of the normal background unused. This raises the question of whether it is necessary to rely on deep models to generate normal backgrounds.

Background-preserving lesion synthesis addresses this issue by starting from normal scans and synthesizing lesions through filling textures into manually generated lesion masks. This approach separates the generation of lesion masks and textures, allowing for finer control over lesions while maintaining the original background structure. Prior work has predominantly relied on heuristic-based methods (Yao et al., 2021; Hu et al., 2023), leveraging the abundance of normal backgrounds to generate lesions of varying sizes and textures at different locations. These studies have demonstrated that the generated data can significantly benefit downstream segmentation tasks.

However, these hand-crafted approaches rely heavily on manual adjustments to ensure that the generated lesions resemble real-world pathology. For example, Hu et al. (2023) manually set grayscale values for texture synthesis, and lesion shapes are generated using morphological operations from

---

[1]For example, a $64^3$ cube crop from a $512^3$ volume occupies $< 0.2\%$ of the total voxels.

ellipsoidal masks. Such hand-crafted rules limits the scalability and generalizability of these methods. In more complex scenarios, such as multi-peak and texture-rich lung nodules or multi-class cardiac lesions, these methods tend to fail (see Sec.4.2). This highlights the need for more flexible and robust data-driven techniques to address these challenges.

A recent study (Chen et al., 2024) uses conditional diffusion (Ho et al., 2020; Rombach et al., 2022) to synthesize abdominal tumors, as illustrated in Fig. 1a, building on background-preserving lesion synthesis . The image is first encoded with VQGAN (Esser et al., 2021), and a latent diffusion (Rombach et al., 2022) learns both the background and lesion, using the lesion mask and background (excluding the lesion) as conditional inputs. However, as the model still needs to generate background, the preservation of background integrity cannot be theoretically guaranteed. While it is possible to fill the area outside the mask with real background data, this may lead to inconsistencies in the final output. From our findings, this approach struggles with high-quality background generation, affecting downstream applications (Sec.4.2), particularly in data-limited scenarios. We also tested a similar image-space conditional diffusion model, which showed similar limitations, though image-space diffusion outperformed latent diffusion due to constraints imposed by the autoencoder.

A few concurrent studies (Lai et al., 2024; Wu et al., 2024; Zhu et al., 2024) have employed advanced generative models for lesion synthesis. Due to differences in research focus or/and the unavailability of their code/models, a comprehensive comparison could not be conducted. Besides, while some studies leverage inpainting techniques to model specific regions of interest (Rouzrokh et al., 2022; Hansen et al., 2024), these methods cannot theoretically guarantee background preservation. Our work emphasizes lesion-specific synthesis rather than broader generative model paradigms, focusing primarily on diffusion-based approaches within the framework of background-preserving lesion synthesis. Comparisons with other generative model paradigms were not included in this study.

## 3 LeFusion: Lesion-Focused Diffusion Model

We propose LeFusion, a lesion-focused diffusion model that concentrates solely on lesion. In Sec.3.1, by combining forward-diffused backgrounds with reverse-diffused foregrounds, LeFusion reconstructs only the lesion, eliminating the need to model complex backgrounds. To address key challenges in lesion texture synthesis (Sec.3.2), LeFusion introduces histogram-based texture control for multi-peak lesions, allowing control over distinct lesion types, and a multi-channel decomposition strategy for multi-class lesions, where classes are generated in separate channels and combined into a single image. In Sec. 3.3, lesion mask diffusion enables control over lesion size, location and boundary, increasing mask diversity and enhancing lesion synthesis quality and flexibility.

### 3.1 Background-Preserving Generation via Inpainting

**Decoupled Lesion and Background Generation.** Inspired by diffusion-based inpainting (Lugmayr et al., 2022; Avrahami et al., 2022), we aim to decouple the generation of lesions and background. As shown in Fig. 2, inpainting predicts the missing parts of an image, particularly lesions. For standard diffusion models (Sohl-Dickstein et al., 2015; Ho et al., 2020), the inference process starts by sampling a noise vector $x_T \sim \mathcal{N}(0, 1)$ and gradually denoising it to produce the output image $x_0$. We focus on 3D images only in this study, while the method can be easily extended to 2D. The original grayscale image is denoted as $\hat{x}_0 \in \mathbb{R}^{D \times H \times W \times 1}$, where $D$, $H$, and $W$ represent the 3D image size. The reverse diffusion step from $x_t$ to $x_{t-1}$ is decoupled into lesion and background components, as shown below. Here, $M_f$ and $M_b$ represent the lesion foreground mask and background mask. The lesion $o_{t-1}$ is predicted through the reverse diffusion process $p_\theta$ using a 3D U-Net, while $\hat{x}_{t-1}$ is derived from $\hat{x}_0$ by adding noise through forward diffusion $q$. $\bar{\alpha}_t$ is defined as $\prod_{s=1}^{t}(1 - \beta_s)$. $\beta_t$ represents the variance schedule parameter, which determines the rate at which noise is gradually injected over time.

$$p_\theta(o_{t-1}|x_t) = \mathcal{N}(o_{t-1}; \mu_\theta(x_t, t), \Sigma_\theta(x_t, t)), \tag{1}$$

$$q(\hat{x}_t|\hat{x}_0) = \mathcal{N}(\hat{x}_t; \sqrt{\bar{\alpha}_t}\hat{x}_0, (1 - \bar{\alpha}_t)\mathbf{I}), \tag{2}$$

$$x_{t-1} = o_{t-1} \odot M_f + \hat{x}_{t-1} \odot M_b. \tag{3}$$

This approach preserves the background accurately without requiring prediction.

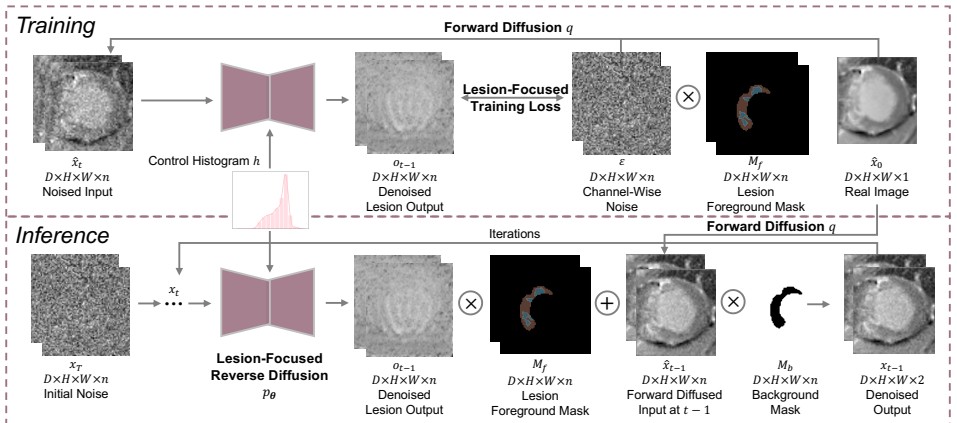

Figure 2: **LeFusion: Lesion-Focused Diffusion Model.** The top illustrates the training process of LeFusion, while the bottom shows the inference. During training, LeFusion avoids learning unnecessary background generation using a lesion-focused loss. In inference, by combining forward-diffused real backgrounds with reverse-diffused generated foregrounds, LeFusion ensures high-quality background generation. Additionally, we introduce histogram-based texture control to handle multi-peak lesions and multi-channel decomposition for multi-class lesions.

**Making Diffusion Model Lesion-Focused.**    The method above, while suitable for all diffusion-based inpainting, cannot guarantee that the denoised output $o_{t-1}$ is focused on the lesion area. Unlike standard diffusion training, where the target region may vary, we know that the part to be inpainted is specifically the lesion in our application. Therefore, we can design the model to predict only the lesion and ignore other regions. To achieve this, we introduce a lesion-focused loss during training. The general diffusion process starts with the original image $\hat{x}_0$, adding Gaussian noise over $T$ time steps (forward diffusion). The neural network is trained to predict the noise distribution at time step $t$ (reverse diffusion), conditioned on the noised image $\hat{x}_t$ and the time step. To ensure the model focuses only on the lesion, a mask $M_f$ is applied, calculating the loss exclusively within the lesion region. The training objective is defined as follows, where $\varepsilon \in \mathbb{R}^{D \times H \times W \times 1}$ represents the noise sampled from a Gaussian distribution:

$$\mathbb{E}_{\hat{x}_0, \epsilon \sim \mathcal{N}(0,1), t} \left[ M_f \| \epsilon - p_\theta \left( \hat{x}_t, t \right) \|_2 \right].  \tag{4}$$

Despite the change in the training objective, inpainting inference remains unaffected. As shown in Eq. 3, outside $M_f$, the predicted lesion $o_{t-1}$ is replaced by the real noised background $\hat{x}_{t-1}$.

## 3.2    FINE CONTROL OF LESION TEXTURES

**Handling Multi-Peak Distributed Lesions.**    In the section above, the model relies on the noised background $\hat{x}_t$ to infer the texture of lesion within the mask. While this works for lesions with minimal texture differences (as with cardiac lesions), it becomes problematic with multi-peak data. As shown in Fig. 3, lung nodules exhibit distinct texture types. We empirically show that relying solely on the background can lead to mode collapse.

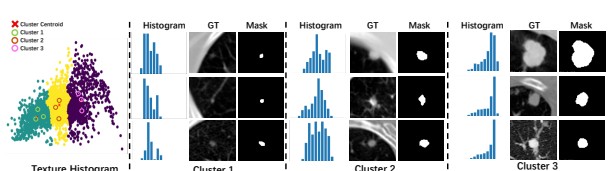

Figure 3: **Illustration of Lung Nodule Texture Histogram Distribution.** Samples are clustered into three groups based on the grayscale image histogram of lesions. The visualized differences between groups are significant, indicating a typical multi-peak distribution. These clusters roughly correspond to ground-glass, part-solid, and solid nodules.

To address this, we propose a simple yet effective approach, histogram-based texture control. The lesion texture histogram $h$ is used as a condition via cross attention Rombach et al. (2022), *i.e.*,

$$o_{t-1} \sim p_\theta \left( \hat{x}_t, h, t \right).  \tag{5}$$

During training, the histogram is computed from the ground truth, and during inference, texture types can be controlled by adjusting the histogram. Notably, this approach requires no additional lesion type annotations, such as nodule attenuation.

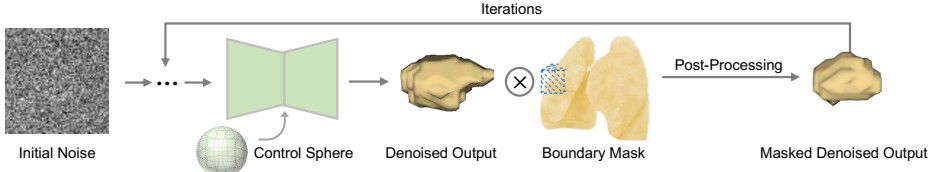

Figure 4: **DiffMask: Lesion Mask Diffusion.** To achieve fine control over lesion size, location, and boundary, we propose two key designs: the boundary mask and the control sphere. The boundary mask removes areas outside the boundary at each diffusion step. The control sphere, trained using the bounding spheres of real masks, enables control over size and location during inference.

In Sec. 4, we demonstrate that the proposed histogram-based texture control is crucial for generating lesions on normal scans. Without it, models tend to fail, biasing towards healthy appearances and producing overly subtle lesions, which degrades the performance of downstream segmentation tasks using these synthetic data.

**Joint Modeling of Multi-Class Lesions.** The method above focuses on single-class lesions, but many medical applications, such as cardiac MRI, require modeling multiple lesion types, like myocardial infarction (MI) and persistent microvascular obstruction (PMO). To capture correlations between lesion types and generate textures for multiple lesions simultaneously, we use a joint modeling strategy called multi-channel decomposition, where each channel corresponds to a different lesion type. The diffusion model generates each lesion in its respective channel, and they are combined through lesion masks.

We expand the input image $\hat{x}_0 \in \mathbb{R}^{D \times H \times W \times 1}$ to $\hat{x}_t \in \mathbb{R}^{D \times H \times W \times n}$, based on the number of lesion classes $n$, where $n = 2$ in the cardiac MRI experiments. Similarly, $\varepsilon \in \mathbb{R}^{D \times H \times W \times n}$ and $o_{t-1} \sim p_\theta(\hat{x}_t, t) \in \mathbb{R}^{D \times H \times W \times n}$ are extended to $n$ channels. The training objective is:

$$\mathbb{E}_{\hat{x}_0, \epsilon \sim \mathcal{N}(0,1), t} \sum_{i=1}^{n} \left[ M_f^{(i)} \left\| \epsilon - p_\theta(\hat{x}_t, t) \right\|_2^{(i)} \right], \qquad (6)$$

where $*^{(i)}$ refers to the channel $i$ of $*$. To combine these channels, we compute $\sum_{i=1}^{n} \left[ M_f^{(i)} o_{t-1}^{(i)} \right]$.

### 3.3 DIVERSIFYING LESION MASKS VIA DIFFMASK

To further enhance the controllability and diversity of lesion generation, we introduce lesion mask diffusion (DiffMask). As shown in Fig. 4, to achieve fine control over lesion size, location, and boundary, we propose two key designs: the boundary mask and the control sphere. The former removes areas outside the boundary at each diffusion step, ensuring the generated mask stays within reasonable limits, while the latter manages the size and location of the lesion. During training, the control sphere is the bounding sphere of a real mask and is concatenated as a condition to the DiffMask input, with the real mask serving as the target. In inference, users can adjust the size and location to generate the desired lesion masks. In terms of implementation, the architecture of DiffMask is similar to the texture diffusion model, also employing multi-channel decomposition to capture shape correlations and spatial distributions between multiple lesion masks. Each output channel is responsible for generating the lesion shape mask of a specific lesion. Finally, we apply a smoothing kernel as a post-processing step.

## 4 EXPERIMENTS

### 4.1 SETUP

**Dataset.** *LIDC: Multi-Peak Lung Nodule CT.* We use LIDC dataset (Armato III et al., 2011), which contains 1,010 chest CT scans, from which 2,624 regions of pathological (**P**) interest (ROIs) corresponding to lung nodules were extracted, along with 135 cases of healthy patients. The dataset was divided into an 808-case training set, comprising 2,104 lung nodule ROIs, and a 202-case test set, containing 520 lung nodule ROIs. Additionally, 3,076 normal (**N**) ROIs were cropped from the 135 healthy patients, representing regions where lung nodules typically appear. These normal

Table 1: **Downstream Lung Nodule Segmentation Dice(%) (↑) and NSD(%) (↑) on LIDC (2011).** P: real pathological cases. P'/N': synthetic pathological cases from pathological/normal cases. N": more synthetic data than N'. **Bold** numbers indicate the best, with red highlighting significantly adverse effects (> 1% lower) compared to the baseline, and blue indicating significantly positive effects (> 1% higher).

| Methods | Training Setting | nnU-Net (2021) | | SwinUNETR (2021) | |
|---|---|---|---|---|---|
| | | Dice (↑) | NSD (↑) | Dice (↑) | NSD (↑) |
| Baseline | P | 78.26 | 88.90 | 78.38 | 88.67 |
| *Texture Synthesis with Real Masks on P'* | | | | | |
| Hand-Crafted (2023) | P+P' | 76.80 | 87.94 | 76.11 | 86.31 |
| Cond-Diffusion (2020; 2022) | P+P' | 77.05 | 87.69 | 77.51 | 88.09 |
| Cond-Diffusion (L) (2024) | P+P' | 76.66 | 87.20 | 76.44 | 86.56 |
| RePaint (2022) | P+P' | 77.57 | 88.07 | 77.14 | 87.96 |
| LeFusion (Ours) | P+P' | 78.77 | 89.25 | 78.43 | 88.54 |
| LeFusion-H (Ours) | P+P' | **80.62** | **90.90** | **80.95** | **90.98** |
| *Texture Synthesis with Hand-Crafted Synthetic Masks (2023) on N'* | | | | | |
| Hand-Crafted (2023) | P+N' | 75.10 | 85.50 | 74.88 | 84.64 |
| Cond-Diffusion (2020; 2022) | P+N' | 76.62 | 86.44 | 76.66 | 87.20 |
| Cond-Diffusion (L) (2024) | P+N' | 76.71 | 86.83 | 77.20 | 87.88 |
| LeFusion (Ours) | P+N' | 77.67 | 87.94 | 77.98 | 88.42 |
| LeFusion-H (Ours) | P+N' | **80.19** | **89.75** | **80.08** | **90.42** |
| *Texture Synthesis with Copied Masks (2023) on N'* | | | | | |
| Copy-Paste | P+N' | 77.29 | 87.60 | 77.80 | 88.84 |
| Hand-Crafted (2023) | P+N' | 76.04 | 86.57 | 76.58 | 87.72 |
| Cond-Diffusion (2020; 2022) | P+N' | 77.00 | 87.68 | 76.68 | 87.40 |
| Cond-Diffusion (L) (2024) | P+N' | 77.15 | 87.83 | 77.38 | 87.51 |
| LeFusion (Ours) | P+N' | 78.49 | 89.22 | 78.55 | 89.06 |
| LeFusion-H (Ours) | P+N' | **81.11** | **91.77** | **81.10** | **91.67** |
| *Enhanced with Diffusion-Based Synthetic Mask (DiffMask)* | | | | | |
| LeFusion-H+DiffMask (Ours) | P+N' | 82.66 | 92.49 | 82.63 | 92.77 |
| LeFusion-H+DiffMask (Ours) | P+N" | 83.19 | 93.21 | 83.07 | 93.10 |
| LeFusion-H+DiffMask (Ours) | P+P'+N" | **83.44** | **93.35** | **83.13** | **93.20** |

ROIs were used for data augmentation in the experiments. *Emidec: Multi-Class Cardiac Lesion MRI.* The Emidec dataset (Lalande et al., 2022) consists of examinations featuring DE-MRI in a short-axis orientation. This dataset offers access to 100 labeled cases, including 33 normal (**N**) and 67 pathological (**P**). The annotations cover 5 classes: background, left ventricle (LV), myocardium (Myo), myocardial infarction (MI), and persistent microvascular obstruction (PMO). We split the 67 **P** cases into 57 for training and 10 for testing. The 57 **P** cases are used to train the data synthesis model. In the downstream evaluation (Sec. 4.2), we use those models to synthesize **P** cases based on both 57 **P** and 33 **N** as the training set.

**Method Comparison.** The following synthesis algorithms are compared with the LeFusion.

*Copy-Paste.* We used the masks from real lesion data and matched them with normal data, copying the original lesion textures onto normal cases to generate new synthetic data.

*Hand-Crafted (Hu et al., 2023).* The lesion mask is represented by the overlapping of multiple ellipsoidal lesion masks, followed by several random morphological operations. The texture is approximated using Gaussian noise and softened through interpolation and Gaussian filtering.

*RePaint (Lugmayr et al., 2022) or Blended-Diffusion Avrahami et al. (2022).* These methods are standard image diffusion models during training, while in inference, they remove and re-fill in the texture within the lesion mask, by combining forward-diffused backgrounds with reverse-diffused foregrounds. The model employs global training loss, which lacks the capability to focus on lesion information. Due to the absence of guidance from lesion category information, it is unable to specify corresponding lesions and is confined to simulating the generation of single-class lesions;

Table 2: **Downstream Cardiac Lesion Segmentation Dice(%) (↑) on Emidec (2022).** The NSD metric is provided in Tab. A1. MI and PMO are [lesion classes. P: real pathological cases. P'/N': synthetic pathological cases from pathological/normal cases. N": more synthetic data than N'. **Bold** numbers indicate the best, with red highlighting significantly adverse effects ($> 1\%$ lower) compared to the baseline, and blue indicating significantly positive effects ($> 1\%$ higher)

| Methods | Training Setting | nnU-Net (2021) | | SwinUNETR (2021) | |
|---|---|---|---|---|---|
| | | MI Dice (↑) | PMO Dice (↑) | MI Dice (↑) | PMO Dice (↑) |
| Baseline | P | 68.61 | 36.32 | 57.79 | 35.76 |
| *Texture Synthesis with Real Masks on P'* | | | | | |
| Hand-Crafted (2023) | P+P' | 69.60 | 36.06 | 57.64 | 34.96 |
| Cond-Diffusion (2020; 2022) | P+P' | 66.89 | 37.76 | 56.75 | 36.31 |
| Cond-Diffusion (L) (2024) | P+P' | 68.07 | 31.93 | 56.97 | 32.72 |
| RePaint (2022) | P+P' | 69.14 | 28.93 | 55.14 | 33.86 |
| LeFusion-S (Ours) | P+P' | 69.88 | 34.79 | 57.85 | 35.63 |
| LeFusion-J (Ours) | P+P' | **69.95** | **38.01** | **59.61** | **37.99** |
| *Texture Synthesis with Hand-Crafted Synthetic Masks (2023) on N'* | | | | | |
| Hand-Crafted (2023) | P+N' | 68.19 | 35.73 | 56.18 | 35.01 |
| Cond-Diffusion (2020; 2022) | P+N' | 67.41 | 31.03 | 56.73 | 35.28 |
| Cond-Diffusion (L) (2024) | P+N' | 67.08 | 36.31 | 56.70 | 33.84 |
| LeFusion-S (Ours) | P+N' | 69.17 | 37.18 | 59.42 | 34.83 |
| LeFusion-J (Ours) | P+N' | **69.87** | **37.31** | **59.56** | **36.19** |
| *Enhanced with Diffusion-Based Synthetic Mask (DiffMask)* | | | | | |
| LeFusion-J+DiffMask (Ours) | P+N' | 69.81 | 40.62 | 58.94 | 39.00 |
| LeFusion-J+DiffMask (Ours) | P+N" | 70.17 | 42.44 | 58.60 | 41.24 |
| LeFusion-J+DiffMask (Ours) | P+P'+N" | 70.34 | **43.54** | **60.54** | 41.70 |
| LeFusion-J-H+DiffMask (Ours) | P+P'+N" | **71.28** | 43.41 | 59.30 | **42.49** |

*Cond-Diffusion (Ho et al., 2020; Rombach et al., 2022).* These methods use the lesion mask and background image information as conditional inputs (Rombach et al., 2022) to a diffusion model (Ho et al., 2020). However, a downside of this approach is that it disrupts the background information. Furthermore, directly using multiple masks as conditional inputs fails to control the corresponding categories, limiting the method to modeling the generation of single-class lesions.

*Cond-Diffusion (L) (Chen et al., 2024).* Cond-Diffusion (L) is conceptually a latent diffusion (Rombach et al., 2022) version of Cond-Diffusion but adds VQGAN (Esser et al., 2021) to map image into latent space for diffusion. For a fair comparison, we fine-tuned open-source code and pre-trained weights by Chen et al. (2024) and used the model outputs directly.

*LeFusion and the Variants (Ours).* Apart from standard LeFusion, there are two variants for fine control of lesion textures. Histogram-Based Texture Control (*-H): A variant of LeFusion that incorporates histogram control information, using the input histogram to guide the generation of multi-peak lesion textures. Multi-Channel Decomposition (*-J): When handling multi-class lesions, standard LeFusion trains individual models *separately* (*-S for distinction), lacking of correlation modeling between classes. LeFusion-J is a generalized version to model multi-class lesions *jointly*.

## 4.2 IMPROVING SEGMENTATION WITH SYNTHETIC DATA

**Lung Nodule Segmentation.** We show that LeFusion can effectively benefit downstream application of training nnUNet (Isensee et al., 2021) and SwinUNETR (Hatamizadeh et al., 2021) to perform lung nodule segmentation. We use the following synthetic subset settings: **P'**: $2104 \times 1$ synthetic ROIs from the 808 real pathological cases **P**; **N'**: $3076 \times 1$ synthetic samples from the 135 normal subjects **N**; **N"**: $3076 \times 2$ synthetic cases from the 135 normal subjects **N**.

Tab. 1 show the Dice and normalized surface distance (NSD). For the first group (Texture Synthesis with Real Masks), the texture of Hand-Crafted (Hu et al., 2023) and RePaint (Lugmayr et al., 2022) differs significantly from the real texture, making it challenging to achieve satisfactory results. Cond-Diffusion (Ho et al., 2020; Rombach et al., 2022) and Cond-Diffusion (L) (Chen et al., 2024), on the other hand, disrupt the background structure of the generated images. Our baseline model, LeFusion, is impacted by the pixel distribution of the background due to the lack of histogram con-

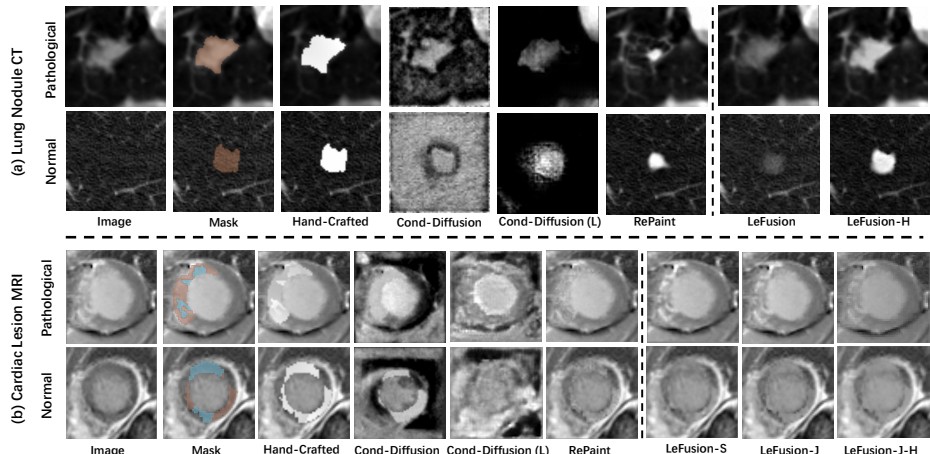

Figure 5: **Visualization of Synthetic Image on Emidec (2022) and LIDC (2011).** We compare the differences in image similarity between synthetic pathological cases generated by different methods, using real pathological cases and using normal regions. More visualizations in Appendix E

trol information, resulting in only a slight improvement over the baseline. We also synthesized lesion data on normal data N using Hand-Crafted Synthetic Masks and masks matched from lesion data, arriving at similar conclusions. In the final set of experiments, we validated the effectiveness of the DiffMask we designed for lesion synthesis and further explored accuracy improvements with increasing amounts of synthetic data. Compared to the baseline, in terms of Dice, we achieved improvements of 5.18% and 4.75% for nnUNet (Isensee et al., 2021) and SwinUNETR (Hatamizadeh et al., 2021), respectively. NSD improvements were 4.4% and 4.53%, respectively.

**Cardiac Lesion Segmentation.** We use the following synthetic subset settings : **P'**: $57 \times 1$ synthetic cases from the 57 real pathological cases **P**; **N'**: $33 \times 2$ synthetic cases from the 33 normal cases **N**; **N''**: $33 \times 5$ synthetic cases from the 33 normal cases **N**. We use combinations of these subsets to train nnUNet (Isensee et al., 2021) and SwinUNETR (Hatamizadeh et al., 2021). The results of the two types of lesions are reported in 2 in terms of the Dice (0-100, higher is better). Due to page limitations, the corresponding NSD table is provided in Appendix Tab. A1.

In the first group of Tab. 2, we apply texture synthesis with real masks. The Hand-Crafted (Hu et al., 2023) produces textures that differ from real textures, leading to a decrease in baseline performance. Cond-Diffusion (Ho et al., 2020; Rombach et al., 2022) and Cond-Diffusion (L) (Chen et al., 2024) disrupts background structure, blurring lesion categories, decreasing MI's performance. RePaint (Lugmayr et al., 2022), focusing on global information, struggles to generate textures that conform to lesion characteristics, resulting in a significant decrease in the Dice for PMO lesions. LeFusion models the two lesions separately, ignoring the correlation between lesions, which leads to improved accuracy for MI but decreased accuracy for PMO lesions. In contrast, our proposed LeFusion-J achieves superior results; For the second group, we expanded the normal data utilizing texture synthesis with hand-crafted synthetic masks. Due to RePaint (Lugmayr et al., 2022)'s inability to distinguish between multiple lesion categories, we did not repeat experiments for it. From the experiments, we observed results similar to those of the first group; For the last, we evaluated our proposed lesion mask synthesis. Our method significantly improved the performance for both MI and PMO. Additionally, as data volume expanded, segmentation Dice consistently improved.

## 4.3 VISUAL QUALITY ASSESSMENT

**Image Quality.** Fig. 5 shows the generation results of different methods for lung nodule CT and cardiac MRI based on lesion images and normal images. We have also quantitatively calculated and compared the paired similarity between our generated images and real images; more details can be found in Appendix D. Fig. 5(a) displays the synthesized visualization of lung nodules (red). The Cond-Diffusion method (Ho et al., 2020; Rombach et al., 2022) and Cond-Diffusion (L) (Chen et al., 2024) disrupt the background structure. The lesions generated by Hand-Crafted (Hu et al., 2023) and RePaint (Lugmayr et al., 2022) fail to capture texture information, such as the grayscale variations characteristic of the lesions. Our baseline LeFusion, without histogram control information, is easily influenced by background features, resulting in the generation of relatively shallow lesions.

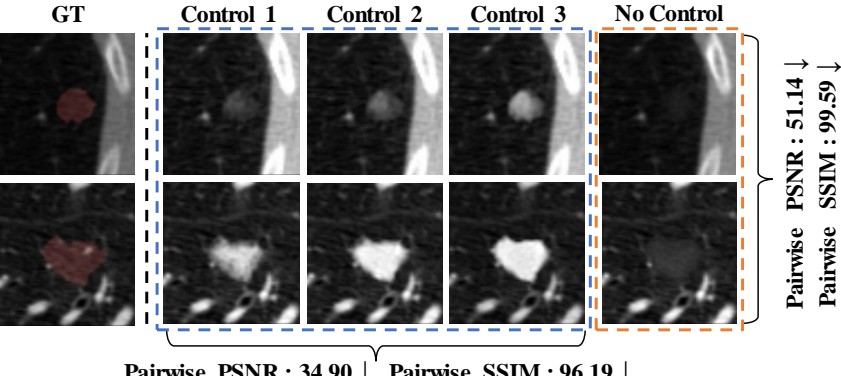

Figure 6: **Illustration of Histograms Control Effectiveness.** GT shows healthy lung tissue with the red area indicating the mask of the generated lesion. Control 1,2, and 3 are images generated under different histogram controls. Lower numbers indicate higher diversity in the generated images.

In contrast, our LeFusion-H can better utilize histogram information to control the generation of lesions. Fig. 5(b) displays the synthesized visualization of two lesions for the heart: MI (blue) and PMO (red). The generation results for the heart show similar conditions. LeFusion-J more accurately captures the textures of the two heart lesions, providing smoother transitions than LeFusion. Due to the small contrast between lesions, LeFusion-J-H yields similar results.

Unpaired perceptual metrics such as FID (Heusel et al., 2017), KID (Bińkowski et al., 2018), and SWD (Karras et al., 2018) primarily assess semantic-level similarity. However, their correlation with visual quality, particularly for medical images, is limited Jayasumana et al. (2024). Despite these shortcomings, we include these metrics as a reference for future work, with detailed results provided in Appendix D. Nevertheless, we believe that evaluating generated data in medical contexts should focus more on the effectiveness of downstream tasks. While assessing synthetic image quality is meaningful, quantitatively comparing different methods remains a challenge.

**Histogram Control Analysis.** We studied the effect of histogram control on the lung nodule dataset. The visualization results are shown in Fig. 6. Under different histogram controls, the attenuation ("transparency") of the generated lesions changes from shallow to deep. Without histogram control information, the generated lung nodules tend to match the pixel distribution of the normal lung background, resulting in overly light lesions. We selected 100 subsets and used LeFusion-H and LeFusion to generate each sample twice, calculating the similarity between pairs using Peak Signal-to-Noise Ratio (PSNR) and Structural Similarity Index Measure (SSIM), lower means more diverse. As shown in the figure, with histogram control, there is greater diversity between samples. We aslo quantitatively analyzed histogram control on lesion areas, revealing shifts in distribution that align well with observations (see Appendix C for details).

**Mask Quality.** We also visualize synthetic lesion masks, as shown in Fig. A1 and Fig. A2. Compared to the hand-crafted masks (Hu et al., 2023), our diffusion-generated masks are closer to the real masks and exhibit a more diverse range of shape patterns. More details can be found in Appendix A.

## 5 CONCLUSION

In conclusion, we introduce LeFusion, a lesion-focused diffusion model that recalibrates learning objectives to lesion areas only. It preserves background by integrating forward-diffused background contexts into the reverse diffusion process. Our methodology is extended to handle challenging multi-peak and multi-class lesions, and further enhanced by a generative model for lesion masks, significantly diversifying our synthetic data. We demonstrate that synthetic data generated by our method can effectively boost the performance of state-of-the-art models.

Data-centric machine learning is increasingly crucial across scientific fields (Reichstein et al., 2019; Rodríguez et al., 2024; Kimanius et al., 2024). This study focuses on generating pathological abnormalities from normal anatomical structures, addressing data bias Mittermaier et al. (2023). While key in medicine, this data synthesis approach is also applicable to other domains with abundant normal data but scarce anomalies, such as industrial, environmental, and material science anomalies.

ETHICS STATEMENT

All experiments in this study were conducted using publicly available datasets. No clinical trials were performed, and the study does not involve human subjects, new dataset releases, or issues related to privacy or security. There are no potential conflicts of interest or sponsorships associated with this work. The paper does not address topics such as discrimination, bias, fairness, legal compliance, or research integrity. However, it is important to acknowledge that our model could potentially be misused to generate fraudulent medical images, such as fabricating lesions in healthy individuals to commit insurance fraud.

REPRODUCIBILITY STATEMENT

To enhance reproducibility, we provide the core code at: `https://github.com/M3DV/LeFusion`. We are committed to fully open-sourcing the code along with the corresponding preprocessed data. Detailed descriptions of data preprocessing, model implementation, experimental settings for our approach and comparison methods (including batch size, learning rate, and checkpoint selection), as well as downstream evaluation procedures, are included in Appendix F. We hope these measures facilitate reproducibility and encourage further research in this area.

ACKNOWLEDGEMENT

This work was supported by a Swiss National Science Foundation grant.

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

## A    MASK VISUALIZATION

Fig. A1 shows the real lung nodule lesion mask (a), the hand-crafted synthetic lung nodule mask (b), and the mask generated by our proposed diffusion model (d). Comparing subfigures (a), (b), and (d) in Fig. A1, it is evident that the real masks exhibit diverse shapes, while the DiffMask-generated masks closely resemble the real ones, also displaying varied forms with similar characteristics. In contrast, the handcrafted masks are relatively uniform in shape and differ significantly from the real masks.

Besides, our proposed DiffMask can control the size and location of the lesion masks, allowing for the generation of more diverse lesions. As shown in Fig. A1 (c) and Fig. A1 (d), we can use the sphere (Size Ball) to precisely control the desired lesion size and its position within the background image. This control enables us to produce various masks.

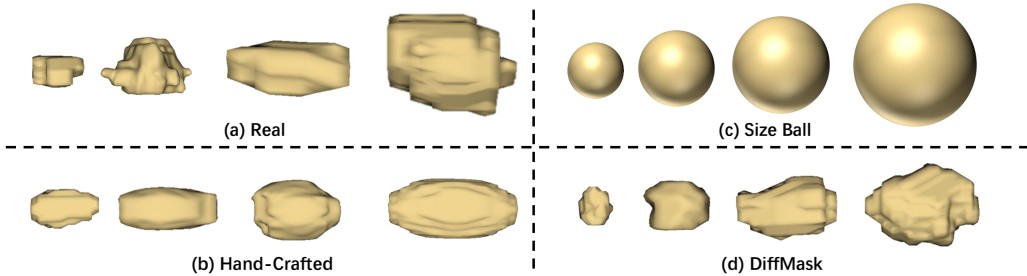

Figure A1: **Visualization of Real / Synthetic Lesion Masks of Lung Nodule.** (a) and (b) show the real masks and hand-crafted masks, (c) displays the control sphere (size ball) used by our proposed DiffMask, and (d) shows the corresponding synthetic mask results generated by DiffMask.

Fig. A2 shows that our diffusion-generated masks are closer to the real masks and exhibit a more diverse range of shape patterns, while Hand-Crafted masks consistently show large, continuous regions.

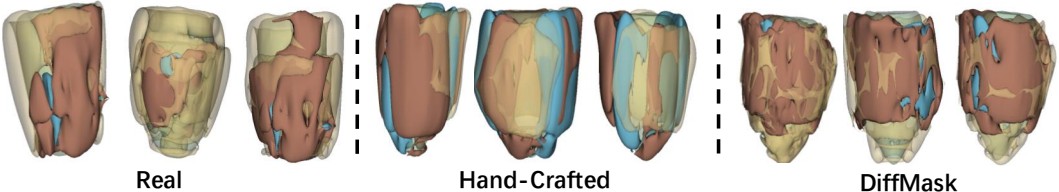

Figure A2: **Visualization of Real / Synthetic Lesion Masks of Cardiac Lesion.**

## B    DOWNSTREAM SEGMENTATION PERFORMANCE

Tab. A1, as a supplement to In Tab. 2, presents the NSD metric measured under the same experimental settings as in In Tab. 2.

In the first set of Tab.A1, we employ texture synthesis with real masks. The Hand-Crafted approach (Hu et al., 2023) generates textures that deviate from real ones, resulting in a decline in baseline performance. Cond-Diffusion (Ho et al., 2020; Rombach et al., 2022) and Cond-Diffusion(L)(Chen et al., 2024) disrupt the background structure, leading to blurring between lesion categories and reducing MI's performance. RePaint (Lugmayr et al., 2022), which emphasizes global information, struggles to produce textures consistent with lesion characteristics, leading to a marked decrease in NSD for PMO lesions. LeFusion-S models the two lesions independently, disregarding the correlation between them, resulting in higher accuracy for MI but reduced accuracy for PMO lesions. Conversely, our proposed LeFusion-J outperforms the other methods; In the second group, we extended the normal data by synthesizing textures using hand-crafted synthetic masks.

Table A1: **Downstream Cardiac Lesion Segmentation NSD(%) (↑) on Emidec (2022).** MI and PMO are lesion classes. P: real pathological cases. P'/N': synthetic pathological cases from pathological/normal cases. N": more synthetic data than N'. **Bold** numbers indicate the best, with red highlighting significantly adverse effects (> 1% lower) compared to the baseline, and blue indicating significantly positive effects (> 1% higher).

| Methods | Training Setting | nnU-Net (2021) | | SwinUNETR (2021) | |
|---|---|---|---|---|---|
| | | MI NSD (↑) | PMO NSD (↑) | MI NSD (↑) | PMO NSD (↑) |
| Baseline | P | 59.27 | 29.19 | 47.66 | 20.51 |
| *Texture Synthesis with Real Masks on P'* | | | | | |
| Hand-Crafted (2023) | P+P' | 60.34 | 35.03 | 47.70 | 19.38 |
| Cond-Diffusion (2020; 2022) | P+P' | 58.21 | 24.04 | 46.35 | 20.87 |
| Cond-Diffusion (L) (2024) | P+P' | 58.98 | 25.96 | 46.83 | 18.72 |
| RePaint (2022) | P+P' | 59.79 | 23.61 | 45.10 | 19.19 |
| LeFusion-S (Ours) | P+P' | **60.77** | 33.13 | 46.66 | 20.05 |
| LeFusion-J (Ours) | P+P' | 60.44 | **36.65** | **48.23** | **24.11** |
| *Texture Synthesis with Hand-Crafted Synthetic Masks (2023) on N'* | | | | | |
| Hand-Crafted (2023) | P+N' | 57.88 | 25.00 | 47.40 | 19.82 |
| Cond-Diffusion (2020; 2022) | P+N' | 58.22 | 22.19 | 46.15 | 20.30 |
| Cond-Diffusion( L) (2024) | P+P' | 58.64 | 26.26 | 47.90 | 19.54 |
| LeFusion-S (Ours) | P+N' | **60.75** | 23.96 | 47.81 | 19.91 |
| LeFusion-J (Ours) | P+N' | 60.68 | **30.62** | **49.69** | **20.71** |
| *Enhanced with Diffusion-Based Synthetic Mask (DiffMask)* | | | | | |
| LeFusion-J+DiffMask (Ours) | P+N' | 61.35 | 38.93 | 48.62 | 22.43 |
| LeFusion-J+DiffMask (Ours) | P+N" | 61.48 | 35.03 | 50.00 | 23.92 |
| LeFusion-J+DiffMask (Ours) | P+P'+N" | 61.27 | **41.62** | **52.82** | 23.77 |
| LeFusion-J-H+DiffMask (Ours) | P+P'+N" | **62.74** | 40.96 | 50.73 | **24.25** |

Due to RePaint (Lugmayr et al., 2022)'s limitations in distinguishing multiple lesion categories, we did not conduct repeated experiments for it. The outcomes were similar to those of the first group; Lastly, we evaluated our proposed lesion mask synthesis. Our approach significantly enhanced the performance for both MI and PMO. Furthermore, as the data volume increased, the downstream segmentation NSD showed consistent improvement.

## C  HISTOGRAM CONTROL ANALYSIS

For a given original image, the histogram information of its lesion region is represented as $I_j$, and the control information as $h_1, h_2, h_3, \ldots, h_n$ (where $i = 1, 2, 3, \ldots, n$), with $n$ being the number of controlled histograms. The resulting output image is defined as $O$, and $O$ can be derived as follows:

$$m_i = (r \times \log l_j + p) + (s \times \log h_i + q) \qquad (7)$$

$$O = e^{m_i}, \quad i = 1, 2, 3, \ldots, n \qquad (8)$$

In this formula, $r$ and $s$ are scaling factors, and $p$ and $q$ are bias offset terms. Based on the above formulas, given the input image and corresponding control information, we can theoretically deduce the histogram of the lesion in the mask area of the generated image.

Fig. A3 shows the impact of histogram control information on the generation of lung nodules in a normal chest background. We randomly sampled 20 cases from the generated lesion data for statistical analysis. The first row represents the controlled histogram, where "No control" indicates that the standard diffusion configuration without histogram control information was used. The second row shows the original mask areas of the 20 cases, as well as the average histogram effects of the generated and predicted data. Rows 3, 4, and 5 display the results of three different sample cases.

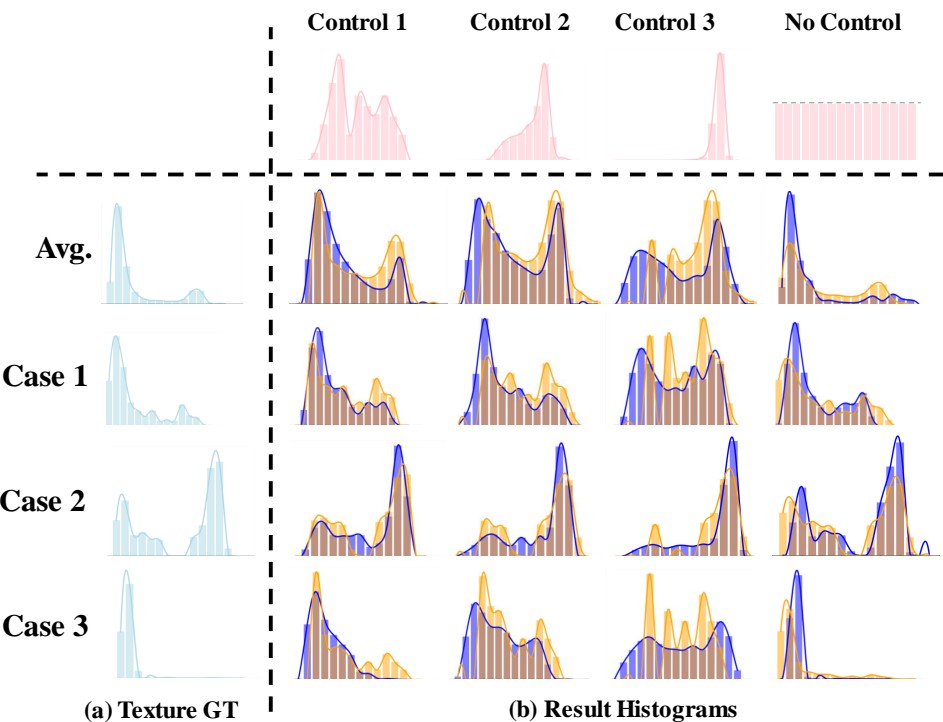

Figure A3: **Illustration of Histograms Control Effectiveness.** The light blue histogram on the far left represents the original mask area. The pink histograms in the first row indicate the control information applied by the diffusion model. The yellow histogram represents the theoretically derived output lesion area, while the darker blue shows the actual generated lesion histogram.

When the model does not use histogram control, the generated histograms of lung nodules in the normal background areas tend to be relatively shallow, influenced by the surrounding background. These histograms resemble the textures of the background mask areas, as shown in the rightmost column. In this column, the dark blue represents the histogram of the image mask area generated by the diffusion model, while the orange represents the theoretically fitted histogram control effect.

After introducing histogram control information, the interaction between the original image lesion areas and the controlled histogram effects causes the generated mask areas to shift increasingly to the right, following the peak effects of the three histograms (control 1, control 2, control 3). The second-row average histogram effect shows the stable trend of this shift.

## D  IMAGE QUALITY EVALUATION

As shown in Tab. A2, we selected Peak Signal-to-Noise Ratio (PSNR) and Structural Similarity Index Measure (SSIM) to calculate the similarity between synthetic pathological cases generated by different methods and real pathological cases. As for cardiac lesions on the EMidec dataset, our proposed LeFusion achieved the highest average PSNR and SSIM. For lymph nodes with only one type of lesion on the LIDC dataset, LeFusion also achieved the highest PSNR. We have also included the Fréchet Inception Distance (FID) (Heusel et al., 2017) and Kernel Inception Distance (KID) (Bińkowski et al., 2018) metrics, as shown in Tab. A3, and the Sliced Wasserstein Distance (SWD) (Karras et al., 2018) in Tab. A4. These results align with the similar phenomena discussed above. These experiments demonstrate that our synthesized lesions are more similar to real lesions across both CT and MRI modalities.

Table A2: **Paired Synthesized Image Quality Assessment of PSNR (↑) and SSIM(%) (↑) on Emidec (2022) and LIDC (2011).** We compare the differences in image similarity between synthetic pathological cases generated by different methods given real pathological cases.

| Methods | Emidec-MI | | Emidec-PMO | | Emidec-Avg. | | LIDC | |
|---|---|---|---|---|---|---|---|---|
| | PSNR ↑ | SSIM ↑ | PSNR ↑ | SSIM ↑ | PSNR ↑ | SSIM ↑ | PSNR ↑ | SSIM ↑ |
| Hand-Crafted (2023) | 9.39 | 8.30 | 7.63 | 9.15 | 8.516 | 8.70 | 1.97 | 0.07 |
| Cond-Diffusion (2020; 2022) | 13.25 | 46.92 | 8.00 | 9.23 | 10.62 | 28.07 | 16.95 | **93.46** |
| Cond-Diffusion(L) (2024) | 14.62 | 69.36 | 12.39 | 61.00 | 13.51 | 65.18 | 15.50 | 90.05 |
| RePaint (2022) | 19.81 | 80.68 | 15.23 | 70.27 | 17.52 | 75.47 | 18.91 | 91.22 |
| LeFusion-S (Ours) | 25.65 | **91.78** | 27.71 | 89.42 | 26.68 | 90.60 | 22.38 | 90.16 |
| LeFusion-J (Ours) | **28.30** | 91.41 | **35.23** | **93.23** | **31.77** | **92.32** | | |

Table A3: **Unpaired Synthesized Image Quality Assessment of FID (%) (↓) and KID (%) (↓) on Emidec (2022) and LIDC (2011).** We compare the differences in image similarity between synthetic pathological cases generated by different methods given real pathological cases.

| Methods | Emidec-MI | | Emidec-PMO | | Emidec-Avg. | | LIDC | |
|---|---|---|---|---|---|---|---|---|
| | FID ↓ | KID ↓ | FID ↓ | KID ↓ | FID ↓ | KID ↓ | FID ↓ | KID ↓ |
| Hand-Crafted (2023) | 19.06 | 3.58 | 17.67 | 16.75 | 18.36 | 10.17 | 12.22 | 2.57 |
| Cond-Diffusion (2020; 2022) | 12.14 | 1.79 | 17.18 | 11.43 | 14.66 | 6.61 | 6.99 | 0.86 |
| Cond-Diffusion(L) (2024) | 12.38 | 1.94 | 22.92 | 9.71 | 17.65 | 5.83 | 9.13 | 1.54 |
| RePaint (2022) | 17.69 | 3.94 | 15.49 | 15.67 | 16.59 | 9.80 | 9.33 | 0.84 |
| LeFusion-S (Ours) | 7.09 | 1.31 | 5.21 | 4.01 | 6.15 | 2.66 | 6.42 | 0.73 |
| LeFusion-J (Ours) | **5.39** | **0.78** | **4.15** | **0.50** | **4.77** | **0.64** | | |

## E    MORE VISUALIZATIONS

In this section, we provide multiple illustrative figures demonstrating the effects of our proposed diffusion model, LeFusion.

**Different Recurrent Length Effects.**    On the one hand, the recurrent operation enhances the stability of texture generation. For instance, employing a medium loop achieves more effective histogram control compared to no recurrent operation, particularly when histogram values are relatively low, as demonstrated in the first and second rows. On the other hand, the recurrent approach improves smoothness and semantic coherence in the intersecting regions between the lesion and the background. For example, in the first row and the first column, the larger loop more effectively captures the spiculated features of the lung nodule, whereas the no recurrent approach tends to overlook the correlation between the lesion and the background.

Table A4:    **Unpaired Synthesized Image Quality Assessment of SWD (1e-4) (↓) on Emidec (2022) and LIDC (2011).** We compare the differences in image similarity between synthetic pathological cases generated by different methods given real pathological cases.

| Methods | Emidec-MI | Emidec-PMO | Emidec-Avg. | LIDC |
|---|---|---|---|---|
| Hand-Crafted (2023) | 26.62 | 4.13 | 15.38 | 10.64 |
| Cond-Diffusion (2020; 2022) | 26.51 | 5.24 | 15.88 | 6.64 |
| Cond-Diffusion (L) (2024) | 15.83 | 5.04 | 10.43 | 7.95 |
| RePaint (2022) | 13.75 | 2.93 | 8.34 | 11.64 |
| LeFusion-S (Ours) | 11.62 | 2.97 | 7.29 | 5.90 |
| LeFusion-J (Ours) | **9.94** | **1.60** | **5.77** | |

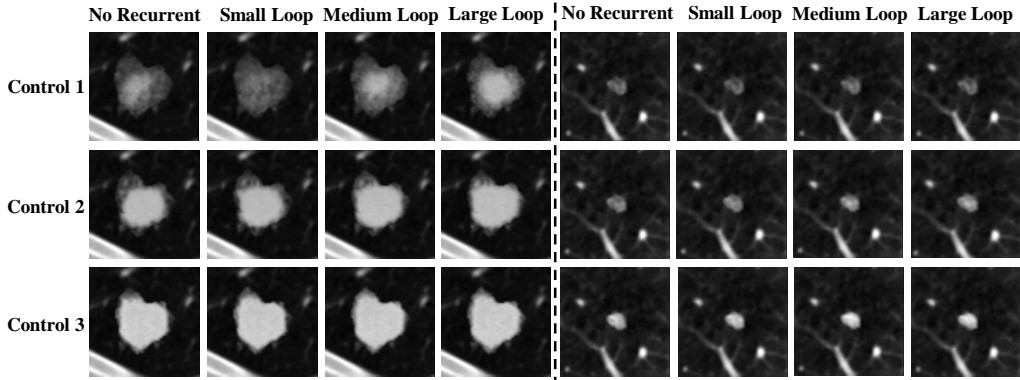

Figure A4: **Visualization of Different Recurrent Length Effects.** "No Recurrent" indicates no recurrence during lesion generation. "Small," "Medium," and "Large" represent varying levels of time step skipping and cyclic folding, ranging from short to long durations. "Control1," "Control2," and "Control3" refer to the generation effects under three different histogram controls.

**Lung Nodule CT.** Fig. A5 presents additional illustrations of the effects of histogram control. The histogram effectively controls the texture of the lesions, while the version without control information tends to generate lighter-colored lung nodules. Fig. A6 provides a visualization of the synthesized lesion results using real samples and their corresponding normal samples.

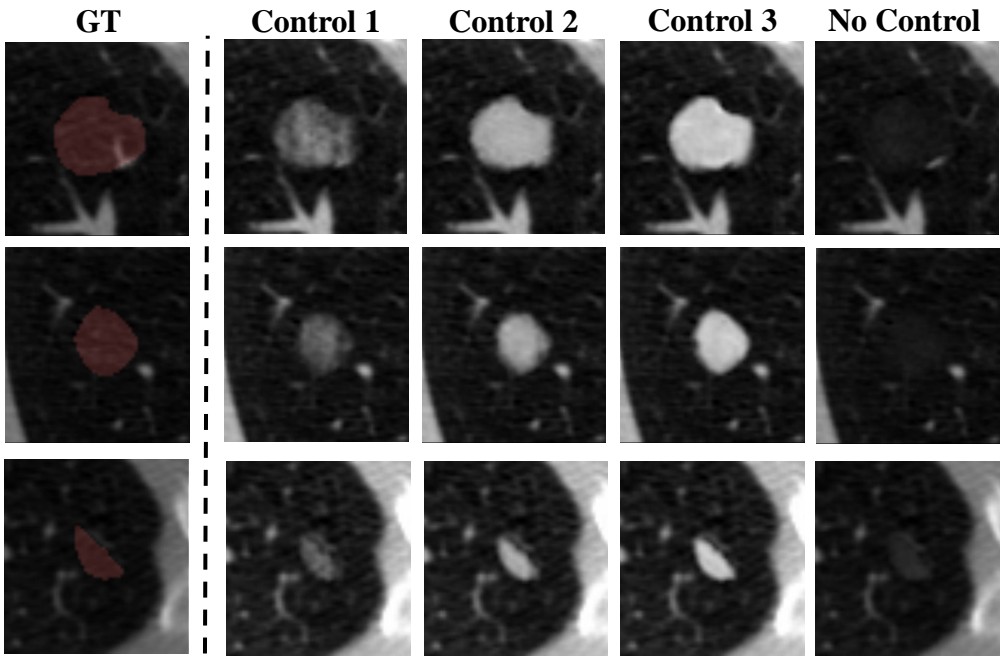

Figure A5: **Illustration of Histograms Control Effectiveness.** GT shows healthy lung tissue with the red area indicating the mask of the generated lesion. Control1, Control2, and Control3 are images generated under three different histogram controls.

**Cardiac Lesion MRI.** Fig. A7 shows the visualization of the denoising process at different stages in LeFusion for inpainting. Fig. A8 and Fig. A9 respectively show the generation of pathological results on lesion cases and normal cases.

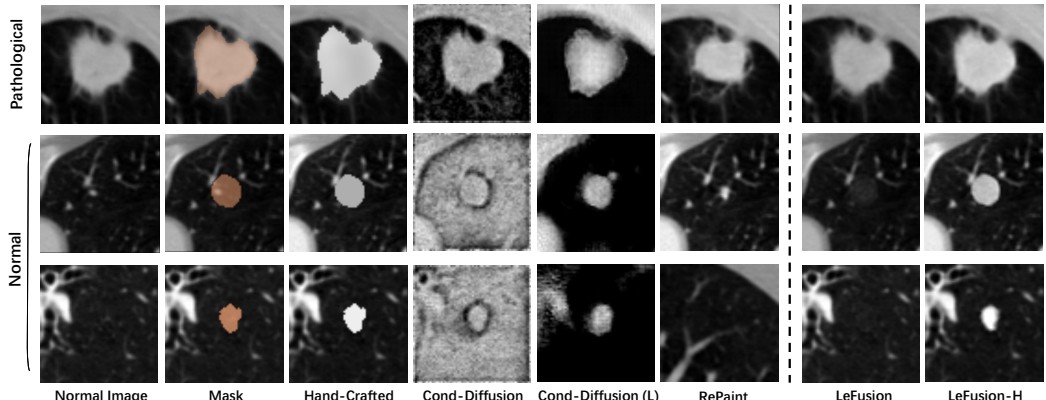

Figure A6: **Visualization of Synthetic Image on LIDC (Armato III et al., 2011).** We compare the differences in image similarity between synthetic cases generated by different methods, using real pathological cases and normal regions.

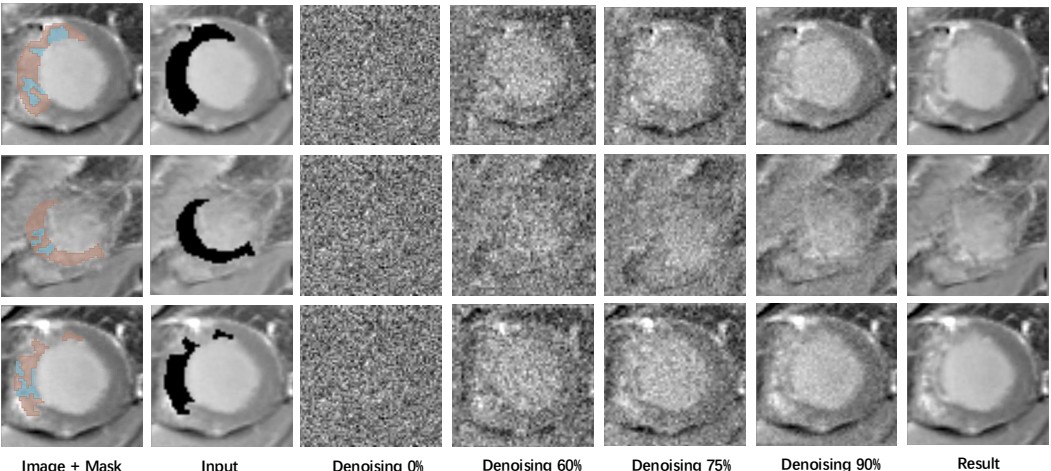

Figure A7: **Visualization of the Denoising Process in LeFusion for Inpainting.** The process is conditioned on the masked input. Starting from a random Gaussian noise sample, the procedure iteratively denoises this input, progressively refining it into a high-quality image with the lesions.

## F    IMPLEMENTATION DETAILS

For the entire experiment, we used 6*A100 (40G) GPUs, including for diffusion and downstream segmentation tasks, with Python 3.8 and PyTorch version 2.4.0.

**Datasets.**    For lung nodules, we followed the common practice of uniformly rescaling the spacing to $1.0 \times 1.0 \times 1.0$ mm  (Han et al., 2019).  Additionally, we uniformly cropped and padded each lesion to a fixed size of $64 \times 64 \times 32$. For cardiac lesions, since the overall data quality was not high and the variation in spacing was minimal, we retained the original spacing to ensure data precision. Furthermore, we uniformly cropped and padded each lesion to a fixed size of $72 \times 72 \times 10$.

**Diffusion Models.**    Training the LeFusion diffusion model for lung nodules requires approximately 4 A100 GPUs for one day. For inference, generating a single data takes about 30-50 seconds on one A100 GPU.

**Recurrent Mechanism for Inpainting.**    To achieve better consistency between lesions and background, we aim to enhance the integration of lesion and background information by allowing the generated images to undergo recurrent processes within the model. Specifically, we adopted some commonly used techniques from the inpainting field (Meng et al., 2022) (Lugmayr et al., 2022).We

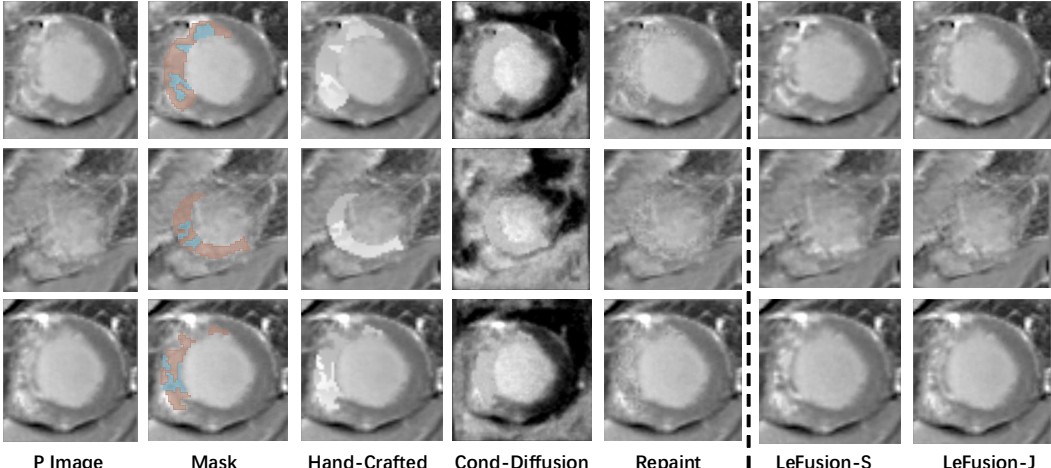

**P Image**    **Mask**    **Hand-Crafted**    **Cond-Diffusion**    **Repaint**    **LeFusion-S**    **LeFusion-J**

Figure A8: **Visualization of Synthetic Images Given Real Lesion Masks on Pathological Cases.**
Our LeFusion are compared with Hand-Crafted Hu et al. (2023), Cond-Diffusion Ho et al. (2020);
Rombach et al. (2022) and RePaint Lugmayr et al. (2022).

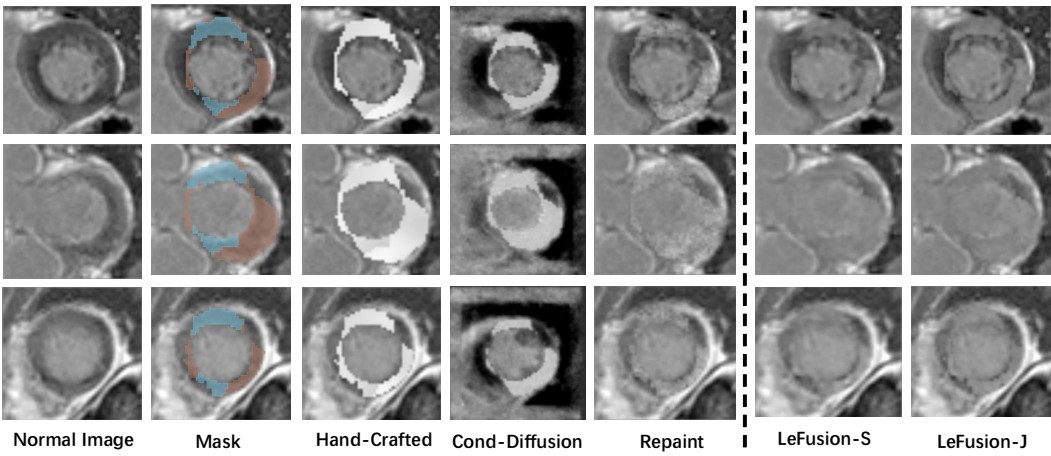

**Normal Image**    **Mask**    **Hand-Crafted**    **Cond-Diffusion**    **Repaint**    **LeFusion-S**    **LeFusion-J**

Figure A9: **Visualization of Synthetic Images Given Synthetic Lesion Masks on Normal Cases.**
Our LeFusion are compared with Hand-Crafted Hu et al. (2023), Cond-Diffusion Ho et al. (2020);
Rombach et al. (2022) and RePaint Lugmayr et al. (2022).

defined two key parameters: the recurrent length and the recurrent sampling frequency. The recurrent length refers to the time span of the recurrent operation, where a larger recurrent length can integrate more information from both the lesion and the background. The recurrent sampling frequency refers to the number of times each recurrent point is sampled. For example, if the initial time step is 300, the recurrent length is set to 2 (skipping two time steps), and the recurrent sampling frequency is 2 (each jump point is revisited once), then the sequence of time steps during the diffusion process would be as follows:

$$\{300, 299, \mathbf{298}, 299, 300, 299, 298, 297, \mathbf{296}, 297, 298, 297, 296, 295, \mathbf{294}, 295, 296, 295, 294, 293...\}$$

**Selection of Histograms.** When generating lesions, the control information for each lesion is randomly selected from three control levels (control3, control2, control1) in the ratio of 75:20:5. We also experimented with other ratios, such as 1:1:1 or 3:2:1, and found that the performance on downstream tasks was roughly the same. To enhance the diversity of the generated lesions, we introduced a fluctuation mechanism for each component of the control information. This mechanism allows the value of each component to randomly vary within ±10% of its original value, while ensuring that the sum of the components remains equal to 1.

**Comparison Method Details.** For the diffusion model architectures compared in our paper—RePaint, Cond-Diffusion, and Cond-Diffusion (L)—all share a similar U-shaped structure. The primary difference between Cond-Diffusion and RePaint lies in their channel configurations, with Cond-Diffusion (L) incorporating latent features as input. In our experiments, we observed that the convergence speed is nearly identical across these models. Therefore, to ensure experimental fairness, we used a unified configuration for all diffusion models. Specifically, all diffusion models were set to 300 timesteps. For both datasets, we adopted a learning rate of 1e-4 and a batch size of 16. To ensure that each diffusion model fully converged, we chose as many training epochs as necessary to ensure the training loss remained stable without continuing to decrease.The training process required approximately 30,000 timesteps for the cardiac dataset and 40,000 timesteps for the LIDC lung nodule dataset.

**Segmentation Models.** We implemented nnUNet (Isensee et al., 2021) and Swin-UNETR (Hatamizadeh et al., 2021) using the MONAI framework. For downstream tasks, both SwinUNETR and nnUNet were trained for 200 epochs. Due to differences in dataset sizes, the training time on a single A100 GPU was approximately 6 to 24 hours for SwinUNETR and 4 to 10 hours for nnUNet.

