# OpenReview forum: "LeFusion: Controllable Pathology Synthesis via Lesion-Focused Diffusion Models"
_ICLR.cc/2025/Conference — ICLR 2025 Spotlight_

### Official Review · Reviewer_f4py · 2024-10-28

**Soundness:** 3
**Presentation:** 3
**Contribution:** 3
**Rating:** 8
**Confidence:** 5

**Summary:**

This paper focuses on generating lesion-containing images from healthy images to address challenges in downstream segmentation tasks, such as real-world data scarcity and long-tail distribution issues. Previous research on medical image synthesis has primarily concentrated on lesion generation design, often overlooking high-fidelity background preservation. The authors propose a lesion-focused diffusion model, LeFusion, which maintains high-fidelity background by integrating the background from forward diffusion into the reverse diffusion process, thus simplifying the learning process and improving output control. Additionally, two effective strategies are introduced: histogram-based texture control and multi-channel decomposition to address the two main challenges in lesion texture synthesis: 1) multimodal and 2) multiclass lesions. The paper is well-written, with comprehensive experimental comparisons.

**Strengths:**

1. The overall paper structure is clear and well-expressed.
2. A novel diffusion model is redesigned from the perspective of high-fidelity background preservation, with two texture generation control techniques developed to address multimodal and multiclass issues.
3. The comparative methods are recent benchmarks from the past two years, making the results highly convincing.

**Weaknesses:**

1.There is a lack of detail on implementation specifics (such as the sampling process) and theoretical support for the method.
2. Analysis and discussion on the continuity at the fusion boundaries between lesion and background are missing, as well as the impact on downstream tasks.

**Questions:**

1. The reverse diffusion sampling process is not clearly defined; it appears to rely solely on the transformation in Equation (1), without detailing the sampling process or providing theoretical justification for omitting it.
2. Although the background from forward diffusion is used as the background in the reverse sampling process, and the loss constraint is applied only to the lesion area, how is continuity and smoothness ensured in the intersecting regions between the lesion and background?

---

> ### Author Response · Authors · 2024-11-25
> **Author Responses (1)**
>
> Thank you for the insightful comments and positive rating. We address your questions as follows:
>
> ---
>
> ### $\bf{Q1: Diffusion \space Formulation}$
>
> > The reverse diffusion sampling process is not clearly defined; it appears to rely solely on the transformation in Equation (1), without detailing the sampling process or providing theoretical justification for omitting it.
>
> $\bf{A:}$
>
> We have added the relevant theoretical derivation in **Section 3.1-Background-Preserving Generation via Inpainting**.
>
> Here, $\hat{x}_0\in\mathbb{R}^{D\times H\times W\times1}$ represents the real image, $M_f$ denotes the lesion foreground mask, and $M_b$ represents the background mask. $\bar{\alpha}_t=\prod _{s=1}^t(1-\beta_s)$ where $\\beta_s$ is the variance schedule, which determines the amount of Gaussian noise added to the data at each time step $t$ . The variance schedule governs the rate at which noise is gradually introduced during the diffusion process.
>
> We employ an unconditional denoising diffusion probabilistic model, expressed as follows:
>
> $$
> p_\theta(o_{t-1}|x_t)=\mathcal{N}(o_{t-1};\mu_\theta(x_t,t),\Sigma_\theta(x_t,t))    \tag{1}
> $$
>
> Since the forward process (Equation 4) is defined as a Markov chain that adds Gaussian noise, we can sample intermediate images $x_t$ at any time step using the following expression:
>
> $$
> q(\hat{x}_t|\hat{x}_0)=\mathcal{N}(\hat{x}_t;\sqrt{\bar{\alpha}_t}\hat{x}_0,(1-\bar{\alpha}_t)\mathbf{I})\tag{2}
> $$
>
> Thus, we can sample the known region $\hat{x}_t \odot M_b$ at any time step $t$. For the unknown region, Equation (1) is used, while for the known region, Equation (2) is applied. This gives us the expression for a reverse step in our method:
>
> $$
> \begin{aligned}x_{t-1}=o_{t-1}\odot M_f+\hat x_{t-1}\odot M_b,o_{t-1}\sim p_\theta\left(x_t,t\right),\hat x_{t-1}\sim q\left(\hat x_0,t\right).\end{aligned}\tag{3}
> $$
>
> Here, $\hat{x} _{t-1}$ is sampled from the given image $\hat{x} _0$ using Equation (2), while $o _{t-1}$ is sampled from the model in Equation (1), based on $x  _t$ from the previous iteration. These two components are then combined using masks to form the new sample $x _{t-1}$.

---

> ### Author Response · Authors · 2024-11-25
> **Author Responses (2)**
>
> ### $\bf{Q2:}$
>
> > Although the background from forward diffusion is used as the background in the reverse sampling process, and the loss constraint is applied only to the lesion area, how is continuity and smoothness ensured in the intersecting regions between the lesion and background?
>
> $\bf{A:}$
>
> Directly pasting lesions onto the background can indeed result in insufficient continuity and smoothness, which negatively impacts downstream tasks. For example, in our Copy-Paste experiments, pasting real lesions directly onto the background without considering their relationship led to a decline in downstream task performance. To address this issue, we sought to iteratively refine the integration of generated lesions and background information within the model, achieving better fusion and enhancing the consistency between lesions and background. Specifically, we adopted commonly used techniques from the inpainting domain [1,2].
>
> To achieve this, we defined two parameters: the recurrent length, which specifies the temporal span of the recurrent operations and allows for better integration of lesion and background information with a longer span, and the recurrent point sampling frequency, which defines the number of repeated sampling iterations at each recurrent point. For instance, if the initial number of timesteps is 300, with a recurrent length of 2 (skipping every two timesteps) and a sampling frequency of 2 (jumping back once at each recurrent point), the **sequence of timesteps** during the diffusion process would be:
>
> $\\{300,299,\textbf{298},299,300,299,298,297,\textbf{296},297,298,297,296,295,\textbf{294},295,296,295,294,293 ...\\}$
>
> In the previous response, we explained how to impose conditional constraints on known regions. As shown in Equation (3), the model predicts $x_{t-1}$ from $x_t$, combining the DDPM output (Equation 1) with samples from the known region. However, during sampling of known pixels using (Equation 3), the model does not consider the rest of the generated image, which can lead to inconsistencies. While the model attempts to reconcile these inconsistencies at each step, it can never fully converge because the same issue arises in subsequent steps.
>
> Additionally, in each reverse step, the variance schedule $\beta_t$ limits the maximum change in the image. This restricted flexibility prevents the model from fully correcting inconsistencies at boundaries in later steps. Consequently, the model requires more time to harmonize the conditional information $\hat{x} _{t-1}$ with the generated information $o _{t-1}$ before proceeding to the next denoising step.
>
> Since DDPMs are trained to generate images within the data distribution, they naturally tend to produce consistent structures. In our resampling approach, we leverage this property of DDPMs to align the model's inputs. Specifically, we diffuse the output $x_{t-1}$ back to $x_t$ using the sampling process defined in:
>
> $$
> q(x_t|x_{t-1})=\mathcal{N}(x_t;\sqrt{1-\beta_t}x_{t-1},\beta_t\mathbf{I})\tag{4}
> $$
>
> Although this operation slightly reduces the sharpness of the output and introduces some noise, certain information from the generated region $o_{t-1}$ is retained in $o_t$. This produces a new $o_t$ that is not only more harmonious with $\hat{x}_t$ but also incorporates its conditional information.
>
> The impact of boundary continuity between lesions and the background on downstream tasks can be observed in **Appendix E: More Visualizations – Different Recurrent Length Effects**.
>
> [1] Meng et al. "SDEdit: Guided Image Synthesis and Editing with Stochastic Differential Equations." ICLR 2022.
>
> [2] Lugmayr et al. "Repaint: Inpainting using denoising diffusion probabilistic models." CVPR 2022

---

> > ### Comment · Reviewer_f4py · 2024-11-26
> >
> > Thank you for your detailed response. I believe my questions have been fully addressed, and I recommend that this paper be accepted for the ICLR conference.

---

> > > ### Author Response · Authors · 2024-11-26
> > > **Thank You for Your Feedback and Recommendation**
> > >
> > > Thank you for your detailed response and for taking the time to carefully review our work. We appreciate your thoughtful feedback and are glad that your questions have been addressed. Please feel free to let us know if there are any additional points we can clarify.

---

### Official Review · Reviewer_XM6f · 2024-10-31

**Soundness:** 3
**Presentation:** 3
**Contribution:** 3
**Rating:** 8
**Confidence:** 5

**Summary:**

This paper introduces a novel 3D lesion inpainting method, LeFusion, which uses diffusion models to address data scarcity in medical imaging. Its primary aim is to generate synthetic lesions in lung CT and cardiac MRI scans for augmenting training data in lesion segmentation tasks. The approach is validated through both visual quality assessments and data augmentation derived segmentation

performance improvement. Three key contributions can be summarised below:
LeFusion Model: The authors identify that existing lesion inpainting methods struggle to preserve anatomically accurate backgrounds alongside the inpainted lesion, remarking that modelling the former is both hard and unnecessary. LeFusion is introduced to address this challenge incorporating two distinct features: (a) Training on a lesion focused diffusion loss, which only considers the lesion region. (b) Preserving the background at inference time with RePaint [1] by generating the lesion separately, while integrating forward-diffused background contexts into the reverse diffusion process. This design yields realistic lesions, better preserved backgrounds and improves data augmentation outcomes in both CT and MRI compared to non-lesion-specific models (Cond-Diffusion) both with and without RePaint based sampling.

Modality-Specific Variants: Two specialized variants are introduced to address modality-specific challenges. LeFusion-H uses histogram-based conditioning to capture diverse lesion textures in CT, succesfully solving the texture mode collapse observed for the baseline LeFusion. LeFusion-J models multiple tissue subtypes in MRI via multi-channel decomposition, which enables the joint generation of different lesion tissue types typically observed in cardiac lesions. Both variants demonstrate superior data augmentation effectiveness in their respective modalities.

DiffMask for Mask Generation: All variants of LeFusion rely on either existing real masks or handcrafted ones as priors for generating lesions in healthy scans. As a more flexible alternative, DiffMask is a diffusion model that generates synthetic lesion masks from basic spatial constraints, defined as a sphere with user specified location and size. Using the generated masks for data augmentation leads to the largest improvement in segmentation performance relative to the baseline in both CT and MRI.

**Strengths:**

Lesion generating models are tools with significant potential for mitigating bias in medical vision AI algorithms concerning lesion detection, segmentation and quantification. Advancements in this topic should be highlighted in venues like this.
The manuscript is sufficiently well written, all the provided Figures/Tables are insightful and adequately formatted.
The choice of a 3D method for this inpainting problem is most adequate for CT and MRI. In these modalities, clinical lesion analysis workflows depend on the visualisation of multiple affected slices and 2D slice-wise inpainting methods would lead to slice-wise discontinuities.

The proposed method is sufficiently contextualised in the Introduction and Related work sections, where the reseach gap is clearly defined. Beyond that, this gap is empirically demonstrated by experimenting with state-of-the-art approaches (Cond-Diffusion variants and RePaint).

The proposed methodologies are thoroughly evaluated through comparisons with multiple other approaches, focusing on visual inspection of inpainted lesions (including comparison with real lesions) and their their downstream usability for training segmentation models. The latter evaluation used two different segmentation models, which contributes to the robustness of the findings across different segmentation training strategies. In addition, evaluating the approach on both MRI and CT datasets, ensures that the findings are not only applicable to one imaging domain.

This paper provides multiple key contributions which not only address the research gap but also deal with modality specific challenges related to lesion texture and shape heterogeneity. The corresponding claims are well supported by the results.

**Weaknesses:**

While S4, the Introduction and Background sections seem to imply that the proposed lesion focused loss is a novel contribution proposed for the first time by the authors. This might not be necessarily true considering that there have been other works that employ similar approaches [2, 3]. While few and perhaps not as thoroughly evaluated, mentioning them could further strengthen the contextualisation of the approach.

The description of the RePaint method in the experimental section implicitly suggests it consists of Cond-Diffusion using the RePaint [1] inference scheme. If that is the case it should be mentioned explicitly, if not then it should be better described.
In the segmentation experiments, it is understood that masks priors for generating lesions in healthy scans (N’) are either derived from real masks, handcrafted or generated by DiffMask. However, additional information should be provided on how exactly the conditioning histograms in this N’ setting are selected when using LeFusion-H variants.

Regarding DiffMask, the definition and role of boundary mask is not very clear. From Figure 4, it is presumed that it corresponds to the bounding box defining the volume crop centred on the lesion. However, the statement “The boundary mask removes areas outside the boundary at each diffusion step” challenges this concept. Further clarity on this point would be appreciated. Furthermore, it is only implicit, that the DiffMask takes the CT/MRI volume crop as an input in addition to the conditioning control sphere. Section 3.3. should be updated to enhance clarity on all these aspects.

Adding supplementary details on how the model training and checkpoint selection was conducted for the RePaint, Cond-Diffusion, Cond-Diffusion (L) would improve transparency.

[Minor]
More detail on the dataset preprocessing would be beneficial for further reproducibility. A mention to the volume resolution is particularly lacking.

The choice of the specific crop-size could be further supported on previous work, for instance [4]. In addition, while not critical for acceptance, it would be interesting to study its effect over the results and would maybe answer the question: “How much local context is it necessary to generate realistic lesion?”

While the purpose of the inpainted lesions is for downstream model training, further validating them using a radiologist would safeguard from potential biases that the generative model might be introducing the lesions.

While describing Tables 1 and 2 it would be useful to clarify what is considered as “significant”. Since no standard deviations were provided, it is implied that these results were obtained for a single fold, so the concept of significance here is vague. In addition, while S5, the robustness of these findings to the specific data split could still be reinforced by adopting some sort of cross validation strategy.
The authors left unclear whether the segmentation model was trained on the volume crops centred on the lesion or on the entire scans. From using the Copy-Paste method in the evaluation, the latter is presumed but it is not explicitly mentioned.

In the cardiac MRI experiments, the LeFusion baseline of modelling the two lesion tissue types with separate models is mentioned as LeFusion in Table 2 but as LeFusion-S in Figure 5 and in the Appendix. It is suggested that the authors stick to one terminology.
As a work mainly focusing on specific diffusion model mechanics for improved lesion inpainting, it makes sense that the evaluation focus on comparing different diffusion based methods. That said, it would still be interesting to see how GAN based approaches like [4, 5] would fair in this comparison.

References:
[1] Lugmayr, Andreas, et al. "Repaint: Inpainting using denoising diffusion probabilistic models." Proceedings of the IEEE/CVF conference on computer vision and pattern recognition. 2022.
[2] Hansen, Colin, et al. "Inpainting Pathology in Lumbar Spine MRI with Latent Diffusion." arXiv preprint arXiv:2406.02477 (2024).
[3] Rouzrokh, Pouria, et al. "Multitask brain tumor inpainting with diffusion models: A methodological report." arXiv preprint arXiv:2210.12113 (2022).
[4] Yang, Jie, et al. "Class-aware adversarial lung nodule synthesis in CT images." 2019 IEEE 16th International Symposium on Biomedical Imaging (ISBI 2019). IEEE, 2019.
[5] Wu, Linshan, et al. "FreeTumor: Advance Tumor Segmentation via Large-Scale Tumor Synthesis." arXiv preprint arXiv:2406.01264 (2024)

**Questions:**

For specific questions please refer to the points made in weaknesses.

---

> ### Author Response · Authors · 2024-11-25
> **Author Responses (1)**
>
> We appreciate your detailed comments and positive rating for the added value of our method in LeFusion. We address your questions as follows:
>
> ---
>
> ### $\bf{Major \space W1: Related \space Loss} $
>
> >While S4, the Introduction and Background sections seem to imply that the proposed lesion focused loss is a novel contribution proposed for the first time by the authors. This might not be necessarily true considering that there have been other works that employ similar approaches [2, 3]. While few and perhaps not as thoroughly evaluated, mentioning them could further strengthen the contextualisation of the approach.
>
> $ \bf{A:} $
>
> We have added the corresponding method [2,3]  references and discussions in the paper to further strengthen the contextualization of lesion focused loss.
>
> [2] Hansen et al. "Inpainting Pathology in Lumbar Spine MRI with Latent Diffusion." arXiv 2024.
>
> [3] Rouzrokh et al. "Multitask brain tumor inpainting with diffusion models: A methodological report." arXiv 2022.
>
> ---
>
> ###  $\bf{Major \space W2: RePaint}$
>
> > The description of the RePaint method in the experimental section implicitly suggests it consists of Cond-Diffusion using the RePaint [1] inference scheme. If that is the case it should be mentioned explicitly, if not then it should be better described.
>
> $ \bf{A:} $
>
> RePaint uses a standard diffusion architecture with repaint mechanism instead of Cond-Diffusion, it incorporates a specific repaint mechanism during the denoising inference process. The standard diffusion differs from the cond-diffusion architecture, particularly in the input(number of channels used as input). Specifically, the Repaint architecture uses only image (1 channel). Cond-Diffusion uses the image, mask and background information (3 channels), while the background information refers to (1-mask)*image. Cond-Diffusion (L) is conceptually a latent diffusion version of Cond-Diffusion but adds VQGAN to map image and background information  into latent space for diffusion, which use image latent features and background information latent features and mask (17 channel). We will clarify the diffusion details further in the paper.
>
> ---
>
> ### $\bf{ Major \space W3 : LeFusion\text{-}H \space Details}$
>
> > However, additional information should be provided on how exactly the conditioning histograms in this N’ setting are selected when using LeFusion-H variants.
>
> $ \bf{A:} $
>
> In the N' setting, our conditioning histograms are randomly selected from three control sources based on the ratio of control3 : control2 : control1 = 75 : 20 : 5. To enhance the diversity of the generated lesions, we introduced a fluctuation mechanism for each component of the control information, allowing its value to randomly vary within ±10% of its original value. Due to space constraints in the main body of the paper, the detailed implementation of the conditional histogram selection method is given in  **Appendix F: Implementation Details - Selection of Histograms**.
>
> ---
>
> ### $\bf{Major \space W4: DiffMask \space Details}$
>
> > Regarding DiffMask, the definition and role of boundary mask is not very clear. From Figure 4, it is presumed that it corresponds to the bounding box defining the volume crop centred on the lesion. However, the statement “The boundary mask removes areas outside the boundary at each diffusion step” challenges this concept. Further clarity on this point would be appreciated.
>
> $ \bf{A:} $
>
> We have added a description addressing this point in the paper. Specifically, for pulmonary nodules, their location should be confined within the thoracic cavity. Therefore, when using the specified control information Control Sphere, if parts of it extend beyond the thoracic boundary, the generated mask may appear outside the thoracic cavity. In such cases, the extraneous parts of the mask should be removed. To ensure the pulmonary nodules appear in anatomically reasonable locations, we use a pre-generated lung mask (https://github.com/jaeho3690/LIDC-IDRI-Preprocessing) during the diffusion process. The method involves removing areas outside the boundary at each diffusion step. We also open source the core code together with our revision.

---

> ### Author Response · Authors · 2024-11-25
> **Author Responses (2)**
>
> ### $\bf{Major \space W5: Volume \space Crop}$
>
> > Furthermore, it is only implicit, that the DiffMask takes the CT/MRI volume crop as an input in addition to the conditioning control sphere. Section 3.3. should be updated to enhance clarity on all these aspects.
>
> $\bf{A:}$
>
> We do not use the CT/MRI volume crop as input to the diffusion model. To generate lesion masks, we rely solely on the corresponding lesion mask—a tensor containing only 0s and 1s, without any additional information such as volume crops—during training. Additionally, we use pre-processed lung masks or heart wall masks.  They also are tensors containing only 0s and 1s, without any volume crop information) to ensure that the generated lesions during inference are restricted to anatomically reasonable locations.
>
> ---
>
> ### $\bf{Major  \space W6: Experiment  \space Details}$
>
> > Adding supplementary details on how the model training and checkpoint selection was conducted for the RePaint, Cond-Diffusion, Cond-Diffusion (L) would improve transparency.
>
> $\bf{A:}$
>
> We have incorporated the corresponding supplementary details into the appendix. For the diffusion model architectures compared in our paper—RePaint, Cond-Diffusion, and Cond-Diffusion (L)—all share a similar U-shaped structure. As discussed above，the primary difference between Cond-Diffusion and RePaint lies in their channel configurations, with Cond-Diffusion (L) incorporating latent features as input. In our experiments, we observed that the convergence speed is nearly identical across these models. Therefore, to ensure experimental fairness, we used a unified configuration for all diffusion models. Specifically, all diffusion models were set to 300 timesteps. For both datasets, we adopted a learning rate of 1e-4 and a batch size of 16. To ensure that each diffusion model fully converged, we chose as many training epochs as necessary to ensure the training loss remained stable without continuing to decrease.The training process required approximately 30,000 timesteps for the cardiac dataset and 40,000 timesteps for the LIDC lung nodule dataset.
>
> ---
>
> ### $\bf{Minor  \space W1: Spacing / Resolution}$
>
> > More detail on the dataset preprocessing would be beneficial for further reproducibility. A mention to the volume resolution is particularly lacking.
>
> $\bf{A:}$
>
> During the preprocessing stage, since the quality of the cardiac data itself was not very high and the variation in spacing was minimal, we did not modify its spacing to ensure data precision. For the LIDC data, due to its large variations in spacing, normalization was necessary. As most studies uniformly rescale the voxels to 1.0 × 1.0 × 1.0 mm [1], we adopted the same approach. Experimentally, we found that spacing had minimal impact on the experimental results, which is consistent with the findings in [2].
>
> We have also mentioned the data resolution in the appendix under the section Implementation Details. Specifically, for the cardiac lesion, we uniformly cropped and padded the size to 72x72x10.For lung nodules, we located each lesion and cropped and padded it to a size of 64x64x32.
>
> We will make all the preprocessed data publicly available, along with the code and pretrained models.
>
> [1] Han et al. "Synthesizing diverse lung nodules wherever massively: 3D multi-conditional GAN-based CT image augmentation for object detection." 3DV 2019.
>
> [2] Yang et al. "AlignShift: bridging the gap of imaging thickness in 3D anisotropic volumes." MICCAI 2020.
>
> ---
>
> ### $\bf{Minor  \space W2: Crop  \space Size}$
>
> > The choice of the specific crop-size could be further supported on previous work, for instance [4]. In addition, while not critical for acceptance, it would be interesting to study its effect over the results and would maybe answer the question: “How much local context is it necessary to generate realistic lesion?
>
> $\bf{A:}$
>
> To achieve better performance in our experiments, we aimed to retain as much information as possible. However, we also had to balance the trade-off between the computational time required for processing and the cropping size, which is constrained by GPU memory limitations. As a result, we cropped the LIDC dataset to 64×64×32 and the EMIDEC dataset to 72×72×10.
>
> We appreciate the reviewer’s suggestion, as exploring the question of "How much local context is necessary to generate realistic lesions?" is indeed an interesting direction. We plan to further investigate this in future work. We have included [4] in the paper, and we will add more relevant references to further support the specific crop size.
>
> [4] Yang et al. "Class-aware adversarial lung nodule synthesis in CT images." ISBI 2019.

---

> ### Author Response · Authors · 2024-11-25
> **Author Responses (3)**
>
> ### $\bf{Minor \space W3: Human \space Evaluation}$
>
> > While the purpose of the inpainted lesions is for downstream model training, further validating them using a radiologist would safeguard from potential biases that the generative model might be introducing the lesions.
>
> $\bf{A:}$
>
> We appreciate the reviewer’s valuable suggestion to involve radiologists in downstream experiments. It is an excellent idea that we intend to pursue.
>
> However, our work primarily focuses on algorithmic innovation and technical feasibility, covering data from two different modalities and mediums: cardiac MRI and pulmonary CT. This distinguishes our study from clinical-oriented research, which often has different focal points.
>
> We have also considered the clinical significance of our work and are currently conducting a clinical-oriented study on lung nodules. In future work, we plan to place greater emphasis on these aspects, including conducting corresponding clinical experiment evaluations.
>
> ---
>
> ### $\bf{Minor \space W4: Table \space Details}$
>
>
> >While describing Tables 1 and 2 it would be useful to clarify what is considered as “significant”. Since no standard deviations were provided, it is implied that these results were obtained for a single fold, so the concept of significance here is vague.
>
> $\bf{A:}$
>
> Thank you for the kind suggestion. We have updated the captions of the relevant tables to make them clearer. Specifically, compared to the baseline (nnU-Net and SwinUNETR), we consider a decrease of 1% in the relevant metric to indicate significant adverse effects, while an increase of 1% signifies significant positive effects.
>
> ---
>
> ### $\bf{Minor  \space W5: Cross  \space Validation}$
>
> > In addition, while S5, the robustness of these findings to the specific data split could still be reinforced by adopting some sort of cross validation strategy.
>
> $\bf{A:}$
>
> We appreciate the suggestion and agree that incorporating a cross-validation strategy would enhance the robustness of the findings. However, due to the computational complexity, it was not feasible to implement cross-validation within the rebuttal period.
>
> Our experiments involve training both generative models and downstream segmentation models under various data and experimental settings. As described in the supplementary materials, one single setup may require days of computation on 4 A100 GPUs. While we recognize the value of cross-validation in improving result stability, the current data split follows standard machine learning practices and is sufficient to support our findings. Moreover, our code and data splits will be made publicly available, ensuring transparency and reproducibility of our results. We hope this adequately addresses your concern.
>
> ---
>
> ### $\bf{Minor \space W6: Downstream \space Details}$
>
> > The authors left unclear whether the segmentation model was trained on the volume crops centred on the lesion or on the entire scans. From using the Copy-Paste method in the evaluation, the latter is presumed but it is not explicitly mentioned.
>
> $\bf{A:}$
>
> For the datasets used in our study, both the diffusion model and the segmentation model were trained using the same preprocessed data. Specifically, for the segmentation model, the EMIDEC dataset utilized entire scans, while the LIDC dataset employed volume crops centered on the lesions.
>
> The diffusion-generated lung nodules and the Copy-Paste experiments were both conducted on the cropped data.It is important to note that not all regions in the cropped normal lung areas are suitable for generating lung nodules; only areas within the thoracic cavity are valid. This necessitated a process of utilizing masks from real lesion data and matching them with normal data.
> We will provide more detailed explanations regarding the specifics of the downstream segmentation experiments in the corresponding sections of the paper.
>
> ---
>
> ### $\bf{Minor \space W7: Typo}$
>
> > In the cardiac MRI experiments, the LeFusion baseline of modelling the two lesion tissue types with separate models is mentioned as LeFusion in Table 2 but as LeFusion-S in Figure 5 and in the Appendix. It is suggested that the authors stick to one terminology.
>
> $\bf{A:}$
>
> We have standardized the terminology and will consistently refer to it as "LeFusion-S."

---

> ### Author Response · Authors · 2024-11-25
> **Author Responses (4)**
>
> ### $\bf{Minor \space W8:}$
>
> > As a work mainly focusing on specific diffusion model mechanics for improved lesion inpainting, it makes sense that the evaluation focus on comparing different diffusion based methods. That said, it would still be interesting to see how GAN based approaches like [4, 5] would fair in this comparison.
>
> $\bf{A:}$
>
> Most recent studies have demonstrated the superiority of diffusion models [4,5], and we have followed this mainstream paradigm. Within this framework, we obtained results similar to those of previous studies: Diffusion models exhibit relatively stable training.
>
> In this paper, however, our primary focus is not on comparing the advantages of GANs and diffusion models or their clinical applications, but rather on addressing the lesion-focused problem. Additionally, since these two works have not been open-sourced, it is challenging to make direct comparisons. Nevertheless, we will include the relevant citations in the **Related Work** section and provide further discussion and detailed comparisons of GAN-based methods.
>
> [4] Yang et al. "Class-aware adversarial lung nodule synthesis in CT images." ISBI 2019.
>
> [5] Wu et al. "FreeTumor: Advance Tumor Segmentation via Large-Scale Tumor Synthesis." arXiv 2024.

---

### Official Review · Reviewer_rF6W · 2024-10-31

**Soundness:** 4
**Presentation:** 4
**Contribution:** 3
**Rating:** 8
**Confidence:** 5

**Summary:**

The authors introduce a latent diffusion model-based method for inserting lesions into healthy medical images while also providing an accompanying mask. They utilize a number of additions to their model to address limitations of prior work or naïve approaches to this task (both pre-existing and seemingly novel), such as combining forward-diffused backgrounds with reverse-diffused foregrounds, introducing intensity histogram-conditioning to the diffusion model to control lesion texture, as well as techniques for further control of the shape, size etc. of the generated lesion. They evaluate their method for a variety of experimental scenarios on 3D cardiac MRI lesion and CT lung nodule generation, showing that their technique results in noticeable improvements to existing approaches with respect to using their generated data to train downstream task segmentation models.

**Strengths:**

Major
1. The paper is polished, well-written and well-presented. Topics and concepts are organized and presented in a digestible fashion.
2. Overall, decent technical novelty. This incorporates many techniques which all come together to result in a strongly-performing methods, some pre-existing (such as combined noised backgrounds with denoised foregrounds), and some seemingly novel (such as histogram-based textural control). Also, despite the many components, the approach still seems relatively watertight because these additions are all pretty lightweight/simple (a good thing). No requirement for an additional network or something of that sort.
3. Overall, results are strong. Clear improvements over baseline methods is basically all cases, using reasonable metrics. They also study a range of training settings, which is good. Clear improvements over Cond-Diffusion, which would be the naïve approach that many would think of first trying for this task; the limitations of it as discussed in the introduction are clear from the experiments.
4. They also have fairly extensive ablation studies for their method, which is important given the number of components that they propose using. There are still a few related questions that I have, but they are minor.
5. In general, the evaluation is fair and appropriate. The datasets are challenging benchmarks, and I think two is sufficient given the wide range of experiments completed on them. There is also a good number of baseline models, especially considering that this task is relatively niche, so the methodological baselines that they compare to seem strong.

Minor
1. The motivation for this problem is clear: pathological subjects are indeed rare, especially for screening populations. Your survey of the limitations of existing lesion synthesis approaches also supports the motivation; for example, they result in low quality backgrounds, they lack precise control over generated lesions, etc.
2. The use of a histogram representation to condition the model on may seem too reductive for some applications, but it seems to work well here (makes sense given the clear correspondence between histogram shape/number of peaks and generated lesion morphology shown in Fig. 3), supported by the clear improvement to your method that including the -H module produced.

**Weaknesses:**

Major
1. Some limitations of impact/scope: This task is clinically important but still fairly niche in medical image analysis, which itself is fairly niche within general machine learning and computer vision. The method (and task itself) also requires that dataset used needs the required annotations, which many medical datasets may not possess, and can be expensive/time-consuming to acquire. Overall, these limit the impact of the work somewhat, in the context of an ML conference at the level of ICLR, compared to a venue a bit more niche like MICCAI.

Minor
1. The benefits from using multi-channel decomposition (comparing the "-J" to no "-J" variants of your model in Table 2) are quite small. Can you provide some analysis or discussion of why this is the case, even if just hypothesizing? (However, I am guessing that the computational requirement to adding this component is practically negligible, so there is not really any harm in including it even if it results in only a very small performance improvement.)
2. You state in the abstract that synthesizing multi-peak and multi-class lesions is a "major challenge" I agree with the multi-peak case given how much your histogram-conditioning improved the generation of such lesions, but based on your channel decomposition module's only very small improvements to performance, I'm unsure if generating multi-class lesions could not already be done well by prior methods. Could you clarify this/point to your results that support this, and/or provide quantitative evidence that multi-class synthesis is challenging for prior approaches?

To summarize, the paper is methodologically solid, with some technical novelty, and demonstrates clear improvements to prior techniques for lesion generation tasks in medical images via well-designed experiments and baselines. However, the main limitation is just that the task is relatively niche within medical image ML, which makes it more niche within general ML, and so may be less impactful at a venue like ICLR as opposed to a medical imaging-focused venue such as MICCAI or MIDL. Still, these limitations do not take away the good things about the paper (of which there are many), so I vote for a marginal accept.

**Questions:**

1. In the tables (e.g. table 1), what do you mean by the significantly adverse/positive effects denoted by red/blue? Could you please clarify this in the text as well via a small note in the table caption(s)?
2. My suggestion: move image quality assessment quantitative results in the appendix (Table A2) to the main text if you have room. These are important metrics. You can shorten the related works to make space, that section doesn't need to be quite so extensive (or some of it could be moved to the supplementary).
    - Also, why didn't you evaluate unpaired perceptual metrics like FID, KID (https://arxiv.org/abs/1801.01401), SWD (https://arxiv.org/abs/1710.10196) etc.? the first two may have limitations for this task given that they use pretrained natural image features, but despite this they are still commonly used metrics for generative medical image models. I would consider adding these for future work, and also explaining why they are not used (particularly for the wider ICLR audience).
3. For the multiclass lesion case/-J model, did you study how performance/generation quality scales with adding more classes? This point may be a bit moot given how small the changes in performance were measured after adding the channel decomposition module to the base model, but I'm still curious.

---

> ### Author Response · Authors · 2024-11-25
> **Author Responses (1)**
>
> We thank you for your detailed comments and positive rating. Please find our point-to-point responses below:
>
> ---
>
> ###  $\bf{Major \space W1 : Study  \space Scope}$
>
> > Some limitations of impact/scope: This task is clinically important but still fairly niche in medical image analysis, which itself is fairly niche within general machine learning and computer vision. The method (and task itself) also requires that dataset used needs the required annotations, which many medical datasets may not possess, and can be expensive/time-consuming to acquire. Overall, these limit the impact of the work somewhat, in the context of an ML conference at the level of ICLR, compared to a venue a bit more niche like MICCAI.
>
> $ \bf{A:} $
>
> Data-centric machine learning is becoming increasingly important across various fields [1,2,3]. We primarily focused on generating pathological abnormalities based on normal anatomical structures (creating abnormal data objects from normal ones). This approach effectively mitigates data bias [4] as we showed.  It is significant in the medical community, but also in a broader context. Our target-oriented data synthesis paradigm is generalizable and can be easily extended to other domains, such as industrial anomaly detection, where normal data is relatively abundant while anomalous data is scarce. Thus, we believe that our approach provides valuable insights in the *AI for Science* domain emphasized by ICLR.
>
> [1] Reichstein et al. "Deep learning and process understanding for data-driven Earth system science." Nature, 2019.
>
> [2] Rodríguez et al. "Machine learning for data-centric epidemic forecasting." Nature Machine Intelligence, 2024
>
> [3] Kimanius et al. "Data-driven regularization lowers the size barrier of cryo-EM structure determination." Nature Methods, 2024.
>
> [4] Mittermaier et al. "Bias in AI-based models for medical applications: challenges and mitigation strategies." NPJ Digital Medicine, 2023.
>
> ---
>
> ### $\bf{Minor  \space W1: Multi\text{-}Channel  \space Decomposition}$
>
> > The benefits from using multi-channel decomposition (comparing the "-J" to no "-J" variants of your model in Table 2) are quite small. Can you provide some analysis or discussion of why this is the case, even if just hypothesizing? (However, I am guessing that the computational requirement to adding this component is practically negligible, so there is not really any harm in including it even if it results in only a very small performance improvement.)
>
> $ \bf{A:} $
>
> Under different settings, the "-J" variants of our model indeed exhibit some performance differences, but they consistently deliver positive improvements. The improvements are particularly noticeable for persistent microvascular obstruction (PMO), a notably long-tailed pathology where only a portion of the dataset contains this condition.
>
> Modeling multiple lesions across different channels to capture their correlations can be considered a promising approach. The relatively small performance improvement in some settings might be due to the weaker correlations between the two lesions. Additionally, the limited amount of cardiac data, even with the inclusion of generated data, may still be insufficient to support robust training for downstream tasks.
>
> It is also important to note that the two cardiac lesions do not exhibit high contrast compared to the background. Consequently, the incorporation of histogram control—the "-H" variants of our model—did not yield significant improvements compared to the no-"H" setting. The "-H" variants are more suitable for clinical applications such as lung nodules, where higher contrast features are present.
>
> ----
>
> ###  $\bf{Minor  \space W2\\&Q3: Multi\text{-}Class}$
>
>
> > - … I'm unsure if generating multi-class lesions could not already be done well by prior methods. Could you clarify this/point to your results that support this, and/or provide quantitative evidence that multi-class synthesis is challenging for prior approaches?
> >
> > - For the multiclass lesion case/-J model, did you study how performance/generation quality scales with adding more classes? …
>
> $ \bf{A:} $
>
> We agree with the reviewer. In some medical datasets, multi-class lesions do objectively exist, and we propose a reasonable solution to address this issue. Our approach demonstrates strong generalizability, poses minimal risk of negatively affecting the diffusion generation process, and introduces almost no additional computational cost.
>
> In the medical field, many labels are often consolidated during annotation, making datasets with a large number of highly correlated classes relatively rare. Therefore, our current design aligns well with a lesion-focused framework. When faced with more complex clinical scenarios, we plan to further extend and refine our approach to multi-class modeling and evaluation.

---

> ### Author Response · Authors · 2024-11-25
> **Author Responses (2)**
>
> ### $\bf{Q1: Table \space Details}$
>
> > In the tables (e.g. table 1), what do you mean by the significantly adverse/positive effects denoted by red/blue? Could you please clarify this in the text as well via a small note in the table caption(s)?
>
> $\bf{A:}$
>
> Thank you for the kind suggestion. We have updated the captions of the relevant tables to make them clearer. Specifically, compared to the baseline (nnU-Net and SwinUNETR), we consider a decrease of one percentage point in the relevant metric to indicate significant adverse effects, while an increase of one percentage point signifies significant positive effects.
>
> ---
>
> ### $\bf{Q2:  Visual \space Quantitative \space  Results}$
>
> > - My suggestion: move image quality assessment quantitative results in the appendix (Table A2) to the main text if you have room. These are important metrics. You can shorten the related works to make space, that section doesn't need to be quite so extensive (or some of it could be moved to the supplementary).
> >
> > - Also, why didn't you evaluate unpaired perceptual metrics like FID, KID (https://arxiv.org/abs/1801.01401), SWD (https://arxiv.org/abs/1710.10196) etc.? the first two may have limitations for this task given that they use pretrained natural image features, but despite this they are still commonly used metrics for generative medical image models. I would consider adding these for future work, and also explaining why they are not used (particularly for the wider ICLR audience
>
> $\bf{A:}$
>
> We emphasize the downstream segmentation results over visual quantitative results, and explain as follows:
>
> Metrics like FID and KID focus primarily on semantic-level similarity, but their alignment with visual quality, especially for medical images, is poor. Consequently, these metrics provide limited guidance when evaluating the fine structural details of medical images. This issue is exacerbated by the lack of pretrained large models specifically tailored for medical imaging. Using natural image-pretrained Inception networks amplifies this problem, as these models emphasize semantic aspects, such as whether a lesion is present, rather than assessing how structurally reasonable the lesion is[1].
>
> Additionally, since the Inception network's pretrained model is designed for 2D RGB images, we are forced to split our 3D medical images into 2D slices for evaluation. This process further disrupts the measurement of 3D structural integrity.
>
> Despite these limitations, we have incorporated the metrics suggested by the reviewer, including FID, KID, and SWD. The detailed results are presented in **Appendix Tables A3 and A4**.
>
> Regarding the "Related Work" section, much of our motivation is discussed there, making it challenging to reduce its length without losing key context.
>
> [1] Jayasumana et al. "Rethinking fid: Towards a better evaluation metric for image generation." CVPR 2024.

---

> ### Author Response · Authors · 2024-11-25
> **Author Responses (3)**
>
> We present an overview of Appendix Tables A3 and A4 below. For a more detailed discussion, please refer to **Appendix D: Image Quality Evaluation**.
>
> **Table A3:** Synthesis Image Quality Assessment of FID(%) (↓) and KID(%) (↓) on Emidec and LIDC . We compare the differences in image similarity between synthetic pathological cases generated by different methods given real patholog-ical cases.
>
> | Methods                  | Emidec-MI FID ↓ | Emidec-MI KID ↓ | Emidec-PMO FID ↓ | Emidec-PMO KID ↓ | Emidec-Avg. FID ↓ | Emidec-Avg. KID ↓ | LIDC FID ↓ | LIDC KID ↓ |
> |-------------------------|-----------------|-----------------|------------------|------------------|-------------------|-------------------|------------|------------|
> | Hand-Crafted      | 19.06           | 3.58            | 17.67            | 16.75            | 18.36             | 10.17             | 12.22      | 2.57       |
> | Cond-Diffusion | 12.14        | 1.79            | 17.18            | 11.43            | 14.66             | 6.61              | 6.99       | 0.86       |
> | Cond-Diffusion (L)| 12.38           | 1.94            | 22.92            | 9.71             | 17.65             | 5.83              | 9.13       | 1.54       |
> | RePaint       | 17.69           | 3.94            | 15.49            | 15.67            | 16.59             | 9.80              | 9.33       | 0.84       |
> | LeFusion-S (Ours)    | 7.09       | 1.31       | 5.21      | 4.01         | 6.15          | 2.66          | **6.42**   |  **0.73**   |
> | LeFusion-J (Ours)   | **5.39**        | **0.78**        | **4.15**         | **0.50**         | **4.77**          | **0.64**          |     —      |     —    |

---

> ### Author Response · Authors · 2024-11-25
> **Author Responses (4)**
>
> **Table A4**: Synthesis Image Quality Assessment of SWD (1e-4) (↓) on Emidec and LIDC . We compare the differences in image similarity between synthetic pathological cases generated by different methods given real pathological cases.
>
> | Methods                  | Emidec-MI ↓     | Emidec-PMO ↓    | Emidec-Avg. ↓    | LIDC ↓           |
> |--------------------------|-----------------|-----------------|------------------|------------------|
> | Hand-Crafted      | 26.62           | 4.13            | 15.38            | 10.64           |
> | Cond-Diffusion  | 26.51        | 5.24            | 15.88            | 6.64            |
> | Cond-Diffusion (L) | 15.83           | 5.04            | 10.43            | 7.95            |
> | RePaint          | 13.75           | 2.93            | 8.34             | 11.64           |
> | LeFusion-S (Ours)   | 11.62      | 2.97        | 7.29        | **5.90**       |
> | LeFusion-J (Ours)   | **9.94**        | **1.60**        | **5.77**         | —       |

---

> ### Comment · Reviewer_rF6W · 2024-11-25
> **Response to author rebuttal**
>
> Thank you for taking the time and effort to respond to my points and suggestions so thoroughly!
>
> As mentioned, my main reason for giving a weak accept instead of an accept was how the application is relatively niche for ICLR. However, you have convinced me of its suitability for the conference, given your points on (1) ICLR emphasizing AI for science/applications work (not to mention, medical image analysis is arguably the largest/most important applied field outside of "standard" computer vision), and (2) evidence that your findings could be useful for other fields.
>
> As such, I'm convinced that this paper should appear in ICLR, and am changing my rating to "accept" in my initial review (as well as changing "contribution" from 2 to 3, following what I described in the preceding paragraph).
>
> Additionally, I take your point for why you don't want to over-emphasize the perceptual metric (FID etc) results, and why it makes sense to keep them in the appendix.

---

> > ### Author Response · Authors · 2024-11-25
> > **Thank You for Feedback and Reconsideration**
> >
> > Thank you for your feedback and for reconsidering our work! We appreciate your recognition of the paper's relevance and your support for its inclusion in ICLR. If there are any remaining aspects that need clarification, please let us know.

---

### Official Review · Reviewer_afPU · 2024-11-04

**Soundness:** 4
**Presentation:** 3
**Contribution:** 3
**Rating:** 8
**Confidence:** 3

**Summary:**

This manuscript presents a diffusion model that utilizes forward-diffused backgrounds and reverse-diffused foregrounds as inputs, allowing the model to concentrate on reconstructing lesions specifically. Additionally, a post-processing method is applied to enhance generation quality.

**Strengths:**

This manuscript is well-motivated, and the experimental results are satisfactory.

**Weaknesses:**

There are several concerns regarding this manuscript:

* The novelty of the proposed approach is limited. The method does not significantly modify the underlying conditional diffusion process but instead introduces variations solely in the input.
* Figure 2 lacks clarity, and it would be beneficial to include the lesion-focused loss in this figure for a more comprehensive understanding.
* The writing lacks organization and is difficult to follow, which may impede readability and comprehension.

**Questions:**

Please revise Figures 1 and 2 to more clearly illustrate the novelty of your proposed approach. Rather than emphasizing the strengths of the paper or incorporating numerous elements into a single pipeline, focus on presenting a straightforward and cohesive pipeline that highlights the mechanisms unique to your method.

---

> ### Author Response · Authors · 2024-11-25
>
> Thank you for your positive feedback and valuable comments. We address your questions as follows:
>
>
> ---
>
> ### $\bf{W1:  Novelty}$
>
> > The novelty of the proposed approach is limited. The method does not significantly modify the underlying conditional diffusion process but instead introduces variations solely in the input.
>
> $\bf{A}$:
>
> We respectfully disagree. Unlike conventional conditional diffusion methods, which introduce variations solely in the input as discussed in the paper, we proposed several improvements. First, as opposed to the standard (conditional) diffusion approach, we preserve high-fidelity backgrounds by integrating *forward-diffused* background contexts into the *reverse diffusion* process. We then modified the training objective to concentrate the diffusion model on lesion textures. Additionally, histogram-based texture control and multi-channel decomposition are proposed to effectively address the challenges of multi-peak and multi-class lesions.
>
> ---
>
> ### $\bf{W2 \\& Question: Figures}$
>
> > - Figure 2 lacks clarity, and it would be beneficial to include the lesion-focused loss in this figure for a more comprehensive understanding.
> >
> > - Please revise Figures 1 and 2 to more clearly illustrate the novelty of your proposed approach. Rather than emphasizing the strengths of the paper or incorporating numerous elements into a single pipeline, focus on presenting a straightforward and cohesive pipeline that highlights the mechanisms unique to your method.
>
>
> $\bf{A}$:
>
> We have revised Fig. 1 and Fig. 2 to enhance their clarity and accessibility. Specifically, we have added detailed annotations, visual highlights, and graphical illustrations to better convey our approach. Fig. 1 has been refined to emphasize “the mechanisms unique to our method”, while Fig. 2 now presents “a more straightforward and cohesive pipeline”.
>
> ---
>
> ### $\bf{W3: Presentation}$
>
>
>
>
> > The writing lacks organization and is difficult to follow, which may impede readability and comprehension.
>
>
> $\bf{A}$:
>
> We understand that our progressive writing style, for example, some methodological motivations are discussed in the **Related Work** section, might lead to different reading experiences depending on the reader’s familiarity with the topic. This style was positively noted by some other reviewers for its clarity in presenting the motivations and context.
> In response to your concerns and to align with feedback from other reviewers, we have made revisions throughout the manuscript, with changes highlighted in blue text. These adjustments aim to improve readability and ensure a smoother flow for readers with diverse backgrounds. We hope these refinements effectively address your concerns.

---

> > ### Comment · Reviewer_afPU · 2024-11-26
> > **Response to author rebuttal**
> >
> > Thanks for your detailed response. I believe that all my concerns have been addressed. Additionally, I have reviewed the responses to other reviewers, and I am confident that the latest version meets the publication standards.

---

> > > ### Author Response · Authors · 2024-11-26
> > > **Thank You for Your Review and Support**
> > >
> > > Thank you for your detailed feedback and for taking the time to review the updates. We’re glad to hear that your concerns have been addressed and that the revised version meets the publication standards. Please don’t hesitate to let us know if there’s anything else we can clarify.

---

### Author Response · Authors · 2024-11-25
**General Response**

We sincerely thank all reviewers for their valuable comments and insightful suggestions. We are encouraged that all reviewers (afPU, rF6W, XM6f, f4py) recognize **strong motivation** and **thorough experiment and analysis** of our research. We are also pleased that the reviewers (rF6W, XM6f, f4py) acknowledge the **novelty** of our model/technical contributions and appreciate the well-structured **presentation** of our paper.

In response to the technical details mentioned by the reviewers, we not only provided explanations in the response and revised paper, but also open source the core code at https://anonymous.4open.science/r/LeFusion. We commit to fully open-sourcing the code along with the corresponding preprocessed data.

We have carefully addressed all the reviewers' concerns in comments and revised the paper. For clarity, we have highlighted the revised parts of the manuscript and supplementary materials in ***blue***. The primary changes are summarized as follows:

1. Added details about technical and experimental methods. (Appendix F), with the source code.

2. Included additional evaluations with unpaired perceptual metrics (Table A3, Table A4).

2. Improved the discussion of related works to provide a more comprehensive comparison.

4. Revised the narrative, figures, and tables along with their corresponding descriptions.

---

### Author Response · Authors · 2024-11-27
**Manuscript Revised and Thank You**

Dear Reviewers,

We sincerely appreciate your insightful feedback. We have carefully addressed all concerns in the revised manuscript, with changes highlighted in blue. Updates include new references, refined content, improved clarity and accuracy in figures, and the addition of an Ethics Statement and a Reproducibility Statement.

As the revision deadline approaches, we welcome any additional feedback and remain open to further discussion on OpenReview.

Thank you again for your thoughtful comments and support in improving this work!

---
Best regards,

LeFusion Authors

---

### Meta-Review · Area_Chair_RSuS · 2024-12-10

**Metareview:**

This paper proposes LeFusion, a lesionfocused diffusion model. By redesigning the diffusion learning objectives to focus on lesion areas, the authors simplify the learning process while preserving high-fidelity backgrounds by integrating forward diffused background contexts into the reverse diffusion process. All reviewers agreed that the paper shall be accepted to this conference.

**Additional Comments On Reviewer Discussion:**

The authors have addressed the concerns raised by the reviewers.

---

### Decision · Program_Chairs · 2025-01-22

Accept (Spotlight)